



Atmospheric
Measurement
Techniques

# Evaluation of single-footprint AIRS CH₄ profile retrieval uncertainties using aircraft profile measurements

Susan S. Kulawik[TS1][1], John R. Worden[2], Vivienne H. Payne[2], Dejian Fu[2], Steven C. Wofsy[3,4], Kathryn McKain[5,6],
Colm Sweeney[5], Bruce C. Daube Jr.[3,4], Alan Lipton[7,☆], Igor Polonsky[7], Yuguang He[7], Karen E. Cady-Pereira[7],
Edward J. Dlugokencky[5], Daniel J. Jacob[4], and Yi Yin[8]

[1]BAER Institute, 625 2nd Street, Suite 209, Petaluma, CA, USA
[2]Jet Propulsion Laboratory, California Institute of Technology, Pasadena, CA, USA
[3]School of Engineering and Applied Sciences and, Harvard University, Cambridge, MA, USA
[4]Department of Earth and Planetary Sciences, Harvard University, Cambridge, MA, USA
[5]National Oceanic and Atmospheric Administration, Global Monitoring Laboratory, Boulder, CO, USA
[6]Cooperative Institute for Research in Environmental Sciences, University of Colorado, Boulder, CO, USA
[7]Atmospheric and Environmental Research, Inc., Lexington, MA, USA
[8]Division of Geological and Planetary Sciences, California Institute of Technology, Pasadena, CA, USA
☆retired

**Correspondence:** Susan S. Kulawik (susan.s.kulawik@nasa.gov)

**Abstract.** We evaluate the uncertainties of methane optimal estimation retrievals from single-footprint thermal infrared observations from the Atmospheric Infrared Sounder (AIRS). These retrievals are primarily sensitive to atmospheric methane in the mid-troposphere through the lower stratosphere ($\sim 2$ to $\sim 17$ km). We compare them to in situ observations made from aircraft during the HIAPER Pole to Pole Observations (HIPPO) and Atmospheric Tomography Mission (ATom) campaigns, and from the NOAA GML aircraft network, between the surface and 5–13 km, across a range of years, latitudes between 60° S to 80° N, and over land and ocean. After a global, pressure-dependent bias correction, we find that the land and ocean have similar biases and that the reported observation error (combined measurement and interference errors) of $\sim 27$ ppb is consistent with the SD between aircraft and individual AIRS observations. A single observation has measurement (noise related) uncertainty of $\sim 17$ ppb, a $\sim 20$ ppb uncertainty from radiative interferences (e.g., from water or temperature), and $\sim 30$ ppb due to "smoothing error", which is partially removed when making comparisons to in situ measurements or models in a way that accounts for this regularization. We estimate a 10 ppb validation uncertainty because the aircraft typically did not measure methane at altitudes where the AIRS measurements have some sensitivity, e.g., the stratosphere, and there is uncertainty in the truth that we validate against. Daily averaging only partly reduces the difference between aircraft and satellite observation, likely because of correlated errors introduced into the retrieval from temperature and water vapor. For example, averaging nine observations only reduces the aircraft–model difference to $\sim 17$ ppb vs. the expected $\sim 10$ ppb. Seasonal averages can reduce this $\sim 17$ ppb uncertainty further to $\sim 10$ ppb, as determined through comparison with NOAA aircraft, likely because uncertainties related to radiative effects of temperature and water vapor are reduced when averaged over a season.

## 1 Introduction

Advances in remote sensing and global transport modeling and an increasingly dense network of surface measurements have led to substantive advances in evaluating the components and error structure of the global methane budget and the processes controlling this budget. For example, Frankenberg et al. (2005, 2011) showed that total column methane

estimates could be derived from near-infrared (NIR) radiances at $\sim 1.6\,\mu m$ measured by the Scanning Imaging Absorption Spectrometer for Atmospheric Cartography (SCIAMACHY). Since then, methane retrievals have also been applied to NIR radiances from the Greenhouse Gases Observing Satellite (GOSAT) instrument (e.g., Parker et al., 2011; Schepers et al., 2012), launched in 2009, and the TROPOspheric Monitoring Instrument (TROPOMI; e.g., Hu et al., 2018), launched in 2017. These data have sufficient accuracy to map regional surface methane enhancements (e.g., Kort et al., 2014; Wecht et al., 2014) and point source anomalies (Varon et al., 2019; Pandey et al., 2019). Estimates of the free-tropospheric methane concentrations from spaceborne measurements in the thermal infrared (TIR) at $\sim 8\,\mu m$ were demonstrated using radiances from the Aura Tropospheric Emission Spectrometer (TES; Worden et al., 2012, 2013b), the Atmospheric Infrared Sounder (AIRS; e.g., Xiong et al., 2013), the Infrared Atmospheric Sounding Interferometer (IASI, e.g., Razavi et al., 2009; De Wachter et al., 2017; Siddans et al., 2017), the Cross-Track Infrared Sounder (CrIS; e.g., Smith and Barnet, 2019), and TIR GOSAT measurements (de Lange and Landgraf, 2018). TIR methane measurements have been used to evaluate the role of fires (e.g., Worden et al., 2013b, 2017a), Asian emissions, and stratospheric intrusions (e.g., Xiong et al., 2009, 2013) in the global methane budget.

The goal of this paper is to evaluate the uncertainties of new methane retrievals from AIRS single-footprint, original (non-cloud-cleared) radiances using aircraft measurements from the HIAPER Pole-to-Pole Observations (HIPPO) and Atmospheric Tomography Mission (ATom) campaigns and National Oceanic and Atmospheric Administration (NOAA) Global Monitoring Laboratory (GML) aircraft network, taken between 2006 and 2017. Evaluation of these uncertainties is needed to determine if AIRS methane data can characterize and improve errors in global chemistry transport models. For example, a recent paper by Zhang et al. (2018) combined synthetic CrIS and TROPOMI methane retrievals and a global inversion system to show that it would be possible to infer the north–south gradient of OH, the primary methane sink, to within 10 %, and temporal variations of OH concentrations. However, knowing the accuracy of the methane data is important for inferring the uncertainty in the spatiotemporal variability of OH. Over decadal timescales, OH can vary by 3 %–5 % (e.g., Turner et al., 2018a, b, 2019; Siddans TS2 et al., 2017). Therefore, to be useful for understanding OH, monthly or seasonally averaged AIRS data should have an uncertainty that is less than 3 %–5 % (55–99 ppb).

In this paper we present an evaluation of methane retrievals derived from AIRS single-footprint radiances. We follow an optimal estimation approach (Rodgers, 2000), based on the heritage of the Aura Tropospheric Emission Spectrometer (TES) algorithm (Bowman et al., 2006), now called the MUlti-SpEctra, MUlti-SpEcies, MUlti-Sensors (MUSES) algorithm (Worden et al., 2006, 2013b; Fu et al.,

2013, 2016, 2018, 2019). The MUSES algorithm uses radiances from one or multiple instruments to quantify and characterize geophysical parameters derivable from those radiances. The optimal estimation method provides the vertical sensitivity (i.e., the averaging kernel matrix) and estimates of the uncertainties due to noise and to radiative interferences such as temperature, $N_2O$, and water vapor. We compare AIRS retrievals with corresponding aircraft data over a range of latitudes and longitudes in order to evaluate the calculated uncertainties over ocean and land. Much of the description of the forward model and retrieval approach is provided in Worden et al. (2012, 2019). We therefore refer the reader to these papers for a more in-depth description of the retrieval approach and only summarize aspects here that are relevant for comparing the AIRS methane retrievals to aircraft data.

## 2 Datasets used in this paper

The quantities of interest that we validate in this paper are (a) the AIRS CH$_4$ dry volume mixing ratio (VMR) at particular pressure values between 750 and 300 hPa or (b) the AIRS CH$_4$ dry VMR partial column XCH$_4$ covering the same pressure range that is measured by the aircraft. We use aircraft profiles which span the pressure range that contains at least 0.20 degrees of freedom for the AIRS CH$_4$ partial column. The retrieval estimates AIRS CH$_4$ dry VMR profile. When a "partial column quantity" is validated, the retrieved CH$_4$ profile is post-processed into partial column XCH$_4$ VMR relative to dry air, with methodology from Connor et al. (2008) and Kulawik et al. (2017), where the VMRs at the pressure levels are weighted according to a pressure weighting function, resulting in a partial column VMR.

### 2.1 Description of AIRS

The AIRS instrument is a nadir-viewing, scanning infrared spectrometer (Aumann et al., 2003; Pagano et al., 2003; Irion et al., 2018; DeSouza-Machado et al., 2018) that is onboard the NASA Aqua satellite and was launched in 2002. AIRS measures the thermal radiance between approximately 3–12 $\mu m$, with a resolving power of approximately 1200. For the 8 $\mu m$ spectral range used for the HDO, H$_2$O, and CH$_4$ retrievals, the spectral resolution is $\sim 1$ wavenumber (cm$^{-1}$), with a gridding of $\sim 0.5$ cm$^{-1}$, and the signal-to-noise (SNR) ranges from $\sim 400$ to $\sim 1000$ over the 8 $\mu m$ region for a typical tropical scene. A single footprint has a diameter of $\sim 15$ km in the nadir; given the $\sim 1250$ km swath, the AIRS instrument can measure nearly the whole globe in a single day. The Aqua satellite is part of the A-Train that consists of multiple satellites and instruments, including TES, in a sun-synchronous orbit at 705 km with an approximately 01:30 and 13:30 Equator crossing time. In this paper, we use only daytime data to match the validation observations.

## 2.2 Overview of aircraft data

Measurements from the HIPPO (Wofsy et al., 2012) and ATom (Wofsy et al., 2018) aircraft campaigns provide excellent datasets for satellite validation, due to their wide latitudinal coverage, the large vertical extent of the profiles (up to 9–12 km), and the availability of campaigns over a wide range of months. Each of the five HIPPO campaigns flew south then north over a period of weeks, often using a different path for the northern and southern legs, with campaign dates in 2009–2011. Atmospheric methane concentrations were measured with a quantum cascade laser spectrometer (QCLS) at 1 Hz frequency, with an accuracy of 1.0 ppb and precision of 0.5 ppb (Santoni et al., 2014). HIPPO methane data are reported on the WMO X2004 scale and have been used in several other studies to evaluate satellite retrievals of methane (e.g., Alvarado et al., 2015; Wecht et al., 2012; Crevoisier et al., 2013). Comparisons with NOAA flask data showed a mean positive bias of 0.85 ppb for the QCLS during the HIPPO campaigns, which is consistent with the estimated QCLS accuracy of 1.0 ppb (Santoni et al., 2014; Kort et al., 2011). We used 396 QCLS CH₄ profiles from the HIPPO campaigns. Using coincidence criteria of ±9 h and ±50 km, 22 271 AIRS observations were processed, of which 5537 passed quality flags. The latitude of the matches ranges from 57° S to 81° N.

We compare AIRS to observations from the ATom aircraft campaigns 1–4 (Wofsy et al., 2018). This comparison provides validation ∼ 7 years after HIPPO, between 2016 and 2018. Similar to HIPPO, these observations include observations in the Pacific Ocean, but ATom also includes observations in the Atlantic (as seen in Table A1 and Fig. 1). ATom methane data are reported on the WMO X2004A scale. We used 289 profiles from the ATom campaigns from the NOAA Picarro instrument (Karion et al., 2013). For more information on the instrument, see https://espo.nasa.gov/sites/default/files/archive_docs/NOAA-Picarro_ATom1234_readme.pdf (last access: 21 December 2020). Using coincidence criteria of ±9 h and ±50 km, 21 225 AIRS observations were processed, of which 4913 passed quality flags. The latitude of the matches ranges from 65° S to 65° N.

The NOAA GML aircraft network observations (Cooperative Global Atmospheric Data Integration Project, 2019) are taken twice per month at fixed sites primarily in North America and also Rarotonga (RTA) at 21° S (Sweeney et al., 2015). NOAA aircraft network methane data are reported on the WMO X2004A scale. Although HIPPO data are not reported on the same scale as ATom and NOAA aircraft network data, differences in values of calibration tanks used for HIPPO (Santoni et al., 2014) on the two different scales are < 1 ppb. We match AIRS and NOAA aircraft observations between 2006 and 2017, with coincidence criteria of 50 km and 9 h, finding ∼ 43 000 matches, and 18 000 good-quality matches following the retrieval, to

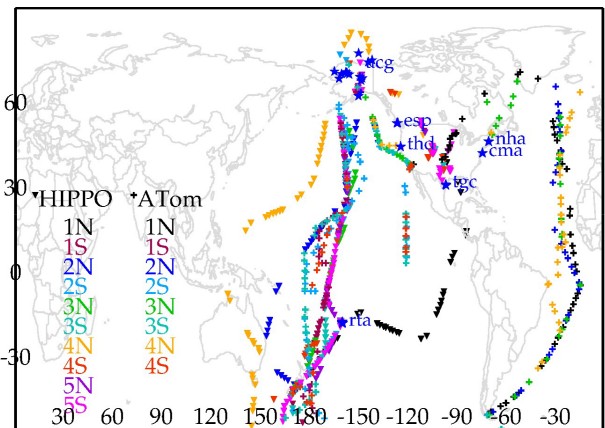

**Figure 1.** Location of aircraft profile measurements used for validation. The upside-down triangles show HIPPO, + symbols show ATom, and blue stars show NOAA ESRL aircraft validation locations.

719 aircraft measurements, at sites ACG (67.7° N, 164.6° E; 401 matches), ESP (49.4° N, 126.5° E; 2743 matches), NHA (43.0° N, 70.6° E; 2682 matches), THD (41.1° N, 124.2° E; 1551 matches), CMA (38.8° N, 74.3° E; 3269 matches), TGC (27.7° N, 96.9° E; 1944 matches), and RTA (21.2° S, 159.8° E; 810 matches).

Figure 1 shows the locations of all the aircraft data used for the comparisons described in this paper. Most of the ocean measurements are from the HIPPO and ATom campaigns that span a range of latitudes, whereas most of the land measurements are taken over North America.

## 3 MUSES-AIRS optimal estimation of CH₄ from single-footprint, original (non-cloud-cleared) AIRS radiances

Worden et al. (2012, 2019) describe in detail the forward model and retrieval approach used for estimating methane from TES and AIRS radiances. The radiative transfer forward model used for this work is the Optimal Spectral Sampling (OSS) fast radiative transfer model (RTM) (Moncet et al., 2008, 2015). In particular, radiances from the thermal infrared bands at 8 and 12 μm are used to quantify profiles of atmospheric concentrations of $CH_4$, HDO, $H_2O$, $N_2O$, as well as temperature, emissivity, and cloud properties. The atmospheric parameters are retrieved as vertical profiles. Since we use optimal estimation, or OE (e.g., Rodgers, 2000; Bowman et al., 2006), to estimate these quantities we can characterize the vertical resolution and uncertainties of these retrievals, which allows us to compare them to models and independent datasets while accounting for the regularization used for the retrieval. We follow the OE approach for the Aura TES instrument (e.g., Bowman et al., 2006; Worden et al., 2006, 2012) but with some differences. First, methane retrievals us-

ing the TES radiances are obtained using only the 8 μm band because of slight calibration differences between the detectors that measure the 12 and 8 μm bands (e.g., Shephard et al., 2008; Connor et al., 2011). For the AIRS retrievals, we use both the 8 and 12 μm bands in order to better constrain temperature in the troposphere and stratosphere. Secondly, the TES-based retrieval uses the ratio of a jointly retrieved N$_2$O profile to the CH$_4$ profile in order to help correct biases related to temperature variations in the upper troposphere–lower stratosphere (UTLS; Worden et al., 2012). However, the N$_2$O correction is not used for the AIRS retrievals because we can jointly estimate temperature in the UTLS region using the 12 μm band. We use similar quality flags as the TES retrievals such as checks on the radiance residual, residual signal, and cloud optical depth (OD) as discussed in Kulawik et al. (2006a, b), except that we screen out cloudy and low-sensitivity cases, resulting in about 1/4 of the data passing screening. Quality flags are discussed in more detail in the Aura-TES user's guide (Herman et al., 2018, pp. 27–30). The specific flags used for AIRS CH$_4$ are as follows, which were set by minimizing the SD of small clusters of retrievals and to standardize the sensitivity.

Here are the recommended cutoffs to select good quality and sensitivity flagging for AIRS CH$_4$:

– The radiance residual rms is $< 1.5$. This parameter is the mean difference between the observed and fit radiance normalized by the radiance measurement error.

– The absolute value of the radiance residual mean is $< 0.15$. This parameter screens off the mean difference of the radiance residual.

– The absolute value of KdotL is $< 0.23$. This parameter is the mean difference of the dot product of the Jacobians and the radiance residual normalized by the radiance measurement error, and smaller values indicate that there is little remaining information in the signal.

– The surface temperature minus the near-surface atmospheric temperature value is $< 30$ K. This ensures that the thermal gradient is less than 30 K between the surface and lowest atmospheric temperature.

– Cloud top pressure is $> 90$ hPa. This ensures that the retrieved cloud top pressure is in or near the troposphere.

– Cloud optical depth is $< 0.3$. This ensures that the cloud is not opaque, and there is fairly uniform sensitivity so that the bias correction is fairly consistent. The bias vs. cloud optical depth is shown in the Supplement.

– Cloud variability vs. wavenumber is $< 1.5 \times$ cloud OD. This ensures that the cloud optical depth does not vary too much over the retrieval window.

– The degrees of freedom are $> 1.1$, defined following Eq. (2). This ensures a minimum sensitivity so that the bias correction is fairly consistent.

– The tropospheric degrees of freedom are $> 0.7$, defined following Eq. (2). This ensures a consistent tropospheric sensitivity, so that the bias correction is fairly consistent.

– The stratospheric degrees of freedom are $< 0.5$, defined following Eq. (2). This ensures that there is a consistent stratospheric sensitivity, so that the bias correction is fairly consistent.

– The predicted error on the column above 750 hPa is $< 53$ ppb. The predicted error is the total error from the linear estimate, Eq. (7b), and is included in the output product. This ensures that the predicted error, which is correlated to the actual error, is not too large.

## 3.1 Retrieval error characteristics

Detailed descriptions of the use of optimal estimation (OE) to infer trace gas profiles from remote sensing radiance measurements' retrieval is included in numerous publications (e.g., Rodgers, 2000; Worden et al., 2006; Bowman et al., 2006). However, we present a partial description here as it is relevant for comparing the AIRS methane retrievals and aircraft profile measurements. As discussed in Rodgers (2000), the estimate for a trace gas profile inferred (or inverted) from a radiance spectrum is described by the following linear equation:

$$\hat{x} = x_a + \mathbf{A}(x - x_a) + \mathbf{G}\mathbf{K}^b b_{\text{error}} + \mathbf{G}n, \tag{1}$$

where $\hat{x}$ is the estimate of log(VMR), $x_a$ is the log of the a priori concentration profile used to regularize the inversion, $\mathbf{G}$ is the gain matrix, and $b_{\text{error}}$ represents errors in systematic parameters, with $\mathbf{K}^b$ the sensitivity of the radiance to changes in $b$. We split $x$ into $[x, y]$, where $x$ is the quantity of interest, the methane profile, and $y$ denotes the jointly estimated quantities (such as temperature, water vapor, clouds, and surface properties), which results in the cross-state error (Worden et al., 2004; Connor et al., 2008).TS4

$$\hat{x} = x_a + \mathbf{A}_{xx}(x - x_a) + \mathbf{G}\mathbf{K}^b b_{\text{error}} + \mathbf{A}_{xy}(y - y_a) + \mathbf{G}n \tag{2}$$

For the AIRS (and TES) OE methane retrievals, $x_a$ comes from the MOZART atmosphere chemistry model (e.g., Brasseur et al., 1998). The vector $x$ is the "true state", or in this case the (log) concentration profile. The matrix $\mathbf{A}$ is the averaging kernel matrix or $\mathbf{A} = \frac{\partial \hat{x}}{\partial x}$ and describes the vertical sensitivity of the measurement. $\mathbf{A}_{xx}$ describes the dependence of $\hat{x}$ on the true state $x$, and $\mathbf{A}_{xy}$ describes the dependence of $\hat{x}$ on the true state $y$, which is non-zero because of correlations in the Jacobians, $\mathbf{K}$, for $x$ and $y$. The matrix $\mathbf{G}$ relates changes in the radiance ($L$ TS5) to perturbations in $x$, $\mathbf{G} = \frac{\partial x}{\partial L}$. The vector $n$ is the noise vector, the matrix $\mathbf{K}$ is the sensitivity of the radiance to changes in (log) concentration $\mathbf{K} = \frac{\partial L}{\partial x} = \frac{\partial L}{\partial \log(\text{VMR})}$, and the set of vectors $b_i$ represent interference errors not estimated from the observed

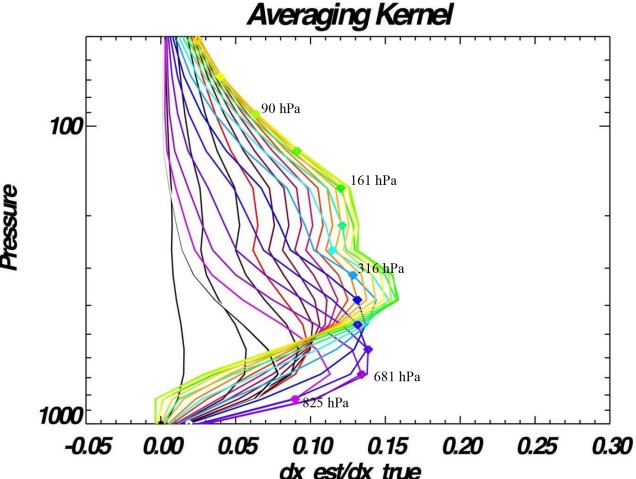

**Figure 2.** The rows of an averaging kernel for CH$_4$ for a tropical scene. The colors help for visualization of the pressure levels for each row of the averaging kernel. The diamonds indicate the pressure level corresponding to the row of the averaging kernel, for pressures 1012, 825, 681, 562, 464, 383, 316, 261, 215, 161, 121, 90, 68, 51 hPa.

radiances. The true state, noise vector, and interference errors as described here are the "true" values and are therefore not actually known but are represented in this form so that we can calculate how their uncertainties affect the estimate $\hat{x}$. An example averaging kernel matrix is shown in Fig. 2 and shows that AIRS-based estimates of methane are most sensitive to methane in the free troposphere and lower stratosphere as demonstrated previously for AIRS and other TIR-based estimates of tropospheric methane (e.g., Xiong et al., 2016; de Lange and Landgraf, 2018).

The degrees of freedom, DOFs, describe the sensitivity of $\hat{x}$ to the true state and are equal to the trace of $\mathbf{A}_{xx}$. The degrees of freedom in the troposphere are equal to the trace of the averaging kernel corresponding to the troposphere, and the degrees of freedom in the stratosphere are equal to the trace of the averaging kernel corresponding to the stratosphere. The troposphere is defined using the tropopause height parameter from version 5 of the NASA Global Modeling and Assimilation Office (GMAO) Goddard Earth Observing System (GEOS-5) model (Molad et al., 2012).

Finally, we look at the quantity of interest, $\hat{x} = \boldsymbol{h}x$. The vector $\boldsymbol{h}$ combines all the necessary operations that map the (log) concentration profiles to whatever quantity is needed such as selecting one particular pressure level (e.g., $\boldsymbol{h} = [0, 0, 0, 1, 0, 0, 0, \ldots]$, selecting a column average, $\boldsymbol{h}$ = pressure weighting function – see Connor et al., 2008, or Kulawik et al., 2017) or selecting the VMR mean (e.g., $\boldsymbol{h}[:] = 1/m$, where $m$ is the number of pressure levels to average).

$$\hat{x} = \boldsymbol{h}\hat{x} \tag{3a}$$

$$\begin{aligned}\hat{x} =&\boldsymbol{h}\boldsymbol{x}_{\mathrm{a}} + \boldsymbol{h}\mathbf{A}_{xx}\left(\boldsymbol{x} - \boldsymbol{x}_{\mathrm{a}}\right) + \boldsymbol{h}\mathbf{G}_x\mathbf{K}^b\boldsymbol{b}_{\mathrm{error}} \\ &+ \boldsymbol{h}\mathbf{A}_{xy}\left(\boldsymbol{y} - \boldsymbol{y}_{\mathrm{a}}\right) + \boldsymbol{h}\mathbf{G}_x\boldsymbol{n}\end{aligned} \tag{3b}$$

In Eq. (3a), the vector $\hat{\boldsymbol{x}}$ (denoted in bold) is converted to the scalar of interest, $\hat{x}$ (non-bold, italic). In our validation comparisons, $\boldsymbol{h}$ is used to select (1) a specific pressure level that is measured by the aircraft, (2) the partial column XCH$_4$ VMR within the pressure levels measured by the aircraft, and (3) the partial column XCH$_4$ between 750 hPa and the top of the atmosphere.

## 3.2 Approach for comparing AIRS measurements to aircraft profiles

A challenge in comparing the satellite-based AIRS measurements to aircraft data is that the aircraft will typically measure only a section of the atmosphere (e.g., the troposphere), whereas the AIRS measurements are sensitive, to varying degrees (see Fig. 2), to the entire atmosphere. To account for these differences, we divide the atmosphere into two parts $\boldsymbol{x} = [\boldsymbol{x}_{\mathrm{c}}, \boldsymbol{x}_{\mathrm{s}}]$, where $\boldsymbol{x}_{\mathrm{c}}$ is the part measured by the aircraft (denoted c for airCraft), and $\boldsymbol{x}_{\mathrm{s}}$ is the part not measured by the aircraft (denoted s for stratospheric): TS6

$$\begin{aligned}\hat{\boldsymbol{x}}_{\mathrm{c}} =&\boldsymbol{h}_{\mathrm{c}}\boldsymbol{x}_{\mathrm{a}} + \boldsymbol{h}_{\mathrm{c}}\mathbf{A}_{\mathrm{cc}}\left(\boldsymbol{x}_{\mathrm{c}} - \boldsymbol{x}_{\mathrm{a}}^{\mathrm{c}}\right) + \boldsymbol{h}\mathbf{G}_{\mathrm{c}}\mathbf{K}^b\boldsymbol{b}_{\mathrm{error}} \\ &+ \boldsymbol{h}_{\mathrm{c}}\mathbf{A}_{cy}\left(\boldsymbol{y} - \boldsymbol{y}_{\mathrm{a}}\right) + \boldsymbol{h}_{\mathrm{c}}\mathbf{A}_{\mathrm{cs}}\left(\boldsymbol{x}_{\mathrm{s}} - \boldsymbol{x}_{\mathrm{a}}^{\mathrm{s}}\right)\mathbf{A}_{\mathrm{cs}}\left(\boldsymbol{x}_{\mathrm{s}} - \boldsymbol{x}_{\mathrm{a}}^{\mathrm{s}}\right) \\ &+ \boldsymbol{h}_{\mathrm{c}}\mathbf{G}_{\mathrm{c}}\boldsymbol{n},\end{aligned} \tag{4}$$

where the term $\mathbf{A}_{\mathrm{cs}}$ is the cross term in the averaging kernel that describes the partial derivatives of the aircraft-measured levels (e.g., the troposphere) to the unmeasured levels (e.g., the stratosphere). Equation (4) describes how the AIRS measurement $\hat{\boldsymbol{x}}_{\mathrm{c}}$ responds to the true state $[\boldsymbol{x}_{\mathrm{c}}, \boldsymbol{x}_{\mathrm{s}}]$. So, if for example, the aircraft measured indices $[0 : 9]$ and did not measure pressure levels $[10 : *]$, then $\mathbf{A}_{\mathrm{cc}} = A[0 : 9, 0 : 9]$ and $\mathbf{A}_{\mathrm{cs}} = A[0 : 9, 10 : 65]$, where $\mathbf{A}$ is the full averaging kernel.

We compare our AIRS observation, $\hat{\boldsymbol{x}}_{\mathrm{c}}$ in Eq. (4), to our aircraft observation, $\boldsymbol{x}_{\mathrm{aircraft}}$. To compare this directly to the aircraft observation (without accounting for AIRS sensitivity), we would compare it to $\hat{\boldsymbol{x}}_{\mathrm{aircraft}}^{\mathrm{c}} = \boldsymbol{h}_{\mathrm{c}}x_{\mathrm{aircraft}}$. The expected total error includes the smoothing error, which is the covariance of the $\boldsymbol{h}_{\mathrm{c}}\mathbf{A}_{\mathrm{cc}}\left(\boldsymbol{x}_{\mathrm{c}} - \boldsymbol{x}_{\mathrm{a}}^{\mathrm{c}}\right)$ (Rodgers, 2000), where the covariance of $\left(\boldsymbol{x}_{\mathrm{c}} - \boldsymbol{x}_{\mathrm{a}}^{\mathrm{c}}\right)$ is the a priori covariance, $\mathbf{S}_{\mathrm{a}}^{xx}$. The smoothing error is as follows:

$$\text{Smoothing error} = \boldsymbol{h}_{\mathrm{c}}\mathbf{A}_{\mathrm{cc}}\mathbf{S}_{\mathrm{a}}^{xx}\mathbf{A}_{\mathrm{cc}}^T\boldsymbol{h}_{\mathrm{c}}^T. \tag{5}$$

We estimate the smoothing error for the partial column XCH$_4$ VMR within the pressure levels measured by the aircraft to be 30 ppb, using Eq. (5). This estimate strongly depends on $\mathbf{S}_{\mathrm{a}}^{xx}$, the a priori covariance, which is the same as in Worden et al. (2012), briefly 5 % diagonal variability with correlations in pressure set from the MOZART model. In Eq. (6a), we apply the AIRS averaging kernel to the aircraft measurement to fully account for the AIRS sensitivity:

$$\hat{\boldsymbol{x}}_{\mathrm{aircraft}}^{\mathrm{c}} = \boldsymbol{h}_{\mathrm{c}}x_{\mathrm{a}} + \boldsymbol{h}_{\mathrm{c}}\mathbf{A}_{\mathrm{cc}}\left(\boldsymbol{x}_{\mathrm{aircraft}}^{\mathrm{c}} - \boldsymbol{x}_{\mathrm{a}}^{\mathrm{c}}\right)$$

$$+ h_\mathrm{c}\mathbf{A}_\mathrm{cs}\left(x_\mathrm{aircraft}^\mathrm{s} - x_\mathrm{a}^\mathrm{s}\right) \tag{6a}$$

$$\hat{x}_\mathrm{aircraft}^\mathrm{c} = h_\mathrm{c} x_\mathrm{a} + h_\mathrm{c}\mathbf{A}_\mathrm{cc}\left(x_\mathrm{aircraft}^\mathrm{c} - x_\mathrm{a}^\mathrm{c}\right). \tag{6b}$$

One issue is that we do not actually have aircraft observations in the "s" part of the atmosphere, $x_\mathrm{aircraft}^\mathrm{s}$, which is used in the second term of Eq. (6a). We have aircraft observations in the "c" part of the atmosphere only, so we apply the averaging kernel to this part of the atmosphere only. Equation (6a) accounts for all of the AIRS smoothing error, whereas Eq. (6b) (the equation used in this work, other than Sect. 3.3) only accounts for the smoothing error from the part of the atmosphere measured by the aircraft profile. The difference from Eqs. (6a) and (6b) is discussed in Sect. 3.3.

Equation (7a) is the predicted bias between $\hat{x}_\mathrm{c}$ (the measured AIRS value) and $\hat{x}_\mathrm{aircraft}^\mathrm{c}$ (the aircraft value with the AIRS averaging kernel applied) and is the expected difference of Eqs. (4) and (6b). Equation (7b) is the covariance of Eq. (7a) and estimates the predicted error:

$$E(\hat{x}_\mathrm{c} - \hat{x}_\mathrm{aircraft}^\mathrm{c}) = h_\mathrm{c}\mathbf{G}_\mathrm{c}\mathbf{K}^b E(b_\mathrm{error}) + h_\mathrm{c}\mathbf{A}_{cy} E(\hat{y} - y_\mathrm{a})$$
$$+ h_\mathrm{c}\mathbf{A}_\mathrm{cs} E(\hat{x}_\mathrm{s} - x_\mathrm{s}) + h_\mathrm{c}\mathbf{G}_\mathrm{c} E(n_\mathrm{error}) \tag{7a}$$

$$E||(\hat{x}_\mathrm{c} - \hat{x}_\mathrm{aircraft}^\mathrm{c})|| = h_\mathrm{c}(\mathbf{G}_\mathrm{c}\mathbf{K}^b\mathbf{S}_\mathrm{a}^{bb}\mathbf{K}_\mathrm{b}^T\mathbf{G}_\mathrm{c}^T + \mathbf{A}_{cy}\mathbf{S}_\mathrm{a}^{yy}\mathbf{A}_{cy}^T$$
$$+ \mathbf{A}_\mathrm{cs}\mathbf{S}_\mathrm{a}^\mathrm{ss}\mathbf{A}_\mathrm{cs}^T + \mathbf{S}_\mathrm{m}^{cc})h_\mathrm{c}^T \tag{7b}$$

Equation (7a) represents the propagation of mean biases from (1) non-retrieved parameters and assumptions, e.g., spectroscopy ($b$); (2) jointly retrieved parameters, e.g., temperature ($y$); (3) "unknown stratospheric true", describing the impact of the part of the atmosphere not covered by the aircraft on the measured part ($x_\mathrm{s}$); or (4) measurement errors ($n$) into biases of $\hat{x}_\mathrm{c}$. The mean bias from Eq. (7a) is difficult to characterize theoretically and is characterized during validation. It is assumed to be primarily from the first term (e.g., spectroscopy). Equation (7b) is the covariance of the terms in 7a, where, e.g., the covariance of $b_\mathrm{error}$ is $\mathbf{S}_\mathrm{a}^{bb}$. Equation (7b) represents the "observation covariance". The square root of Eq. (7b) is the predicted observation error. Although Eq. (7b) has overall zero bias, it can produce regional and temporal biases, e.g., as seen in Connor et al. (2016), where these biases approach zero over long enough spatial or temporal scales. The error covariances all represent fractional errors, in log(VMR). Because of the retrieved quantity log(VMR), the error in ppb is approximately the fractional error times the methane value in ppb.

For the purpose of evaluating the AIRS methane measurement uncertainties and comparing the AIRS methane to aircraft in situ measurements, we refer to the four terms on the right side of Eq. (7b) as follows:

1. $\mathbf{S}_\mathrm{b}^{cc}$ is the systematic error due to terms that are not accounted for in the retrieval state vector, such as spectroscopy and calibration; these terms are estimated by comparisons with the aircraft data. A pressure-dependent bias correction, described in Sect. 3.4, of approximately $-60$ ppb is used to correct this systematic bias.

2. $\mathbf{A}_{cy}\mathbf{S}_\mathrm{a}^{yy}\mathbf{A}_{cy}^T$ is the cross state, which is included in the MUSES-AIRS methane estimate product files and is the propagation of temperature, water vapor, and cloud errors into AIRS. The errors in the retrieved temperature and water vapor at nearby location are correlated over short spatiotemporal scales, as described in Sect. 4, and so this error does not reduce with averaging nearby observations. However, monthly or seasonal averages reduce the cross-state error because systematic errors from temperature, water, or cloud can be assumed to vary pseudo-randomly over larger timescales.

3. $\mathbf{A}_\mathrm{cs}\mathbf{S}_\mathrm{a}^\mathrm{ss}\mathbf{A}_\mathrm{cs}^T$ is the "validation uncertainty" due to knowledge uncertainty of the stratosphere, although this may also contain other levels that are also not measured by the aircraft. This is the smoothing error which cannot be removed from the comparisons because the aircraft does not make measurements at the "s" ("stratospheric") levels. We estimate this validation uncertainty to be $\sim 10$ ppb (estimated in Sect. 3.3). This estimate depends on the accuracy of the model used to extend the aircraft profile during the validation process and was estimated for the model that we used in validation.

4. $\mathbf{S}_\mathrm{m}^{cc}$ is the measurement error, which is included in the AIRS methane estimate product files. The measurement error is random and is expected to reduce as the inverse square root of the number of observations averaged. We estimate this error to be $\sim 18$ ppb (using the last term of Eq. 7b) and shown in Fig. 3) and find it to be a random error that reduces with averaging.

Figure 3 shows the predicted errors for the AIRS partial column XCH$_4$ VMR within the pressure levels measured by the aircraft. The measurement error (light green) is 18 ppb (from the last term of Eq. 7b), and the total error for a single observation (including smoothing error) is 41 ppb. A component of the total error, the cross-state error, is estimated to be 21 ppb (from Eq. 7b).

### 3.3 Estimating validation uncertainty due to aircraft not measuring the stratosphere

A typical aircraft profile will only measure part of the troposphere and rarely measure into the stratosphere. However, the AIRS methane profile measurements are sensitive to methane variations over the whole atmosphere, as shown by the averaging kernel matrix in Fig. 2. Similarly, the true state in the troposphere influences retrieved values in the stratosphere. Options for dealing with this are (a) extending the true profile with the AIRS prior or (b) extending the true profile with a model profile value.

This section estimates this uncertainty by calculating the difference of $x_\mathrm{aircraft}^\mathrm{c}$ for Eq. (6a) minus Eq. (6b) when extending the aircraft using two different "true" profiles taken from two different global atmospheric chemistry models,

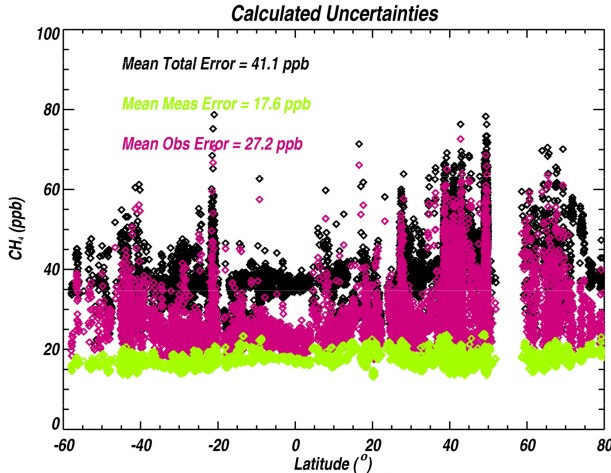

**Figure 3.** Calculated errors for AIRS measurements shown in this paper. The total error shown is the smoothing error (Eq. 5) plus the observation error (Eq. 7b). The measurement error is the last term of Eq. (7b) and the only fully random error.

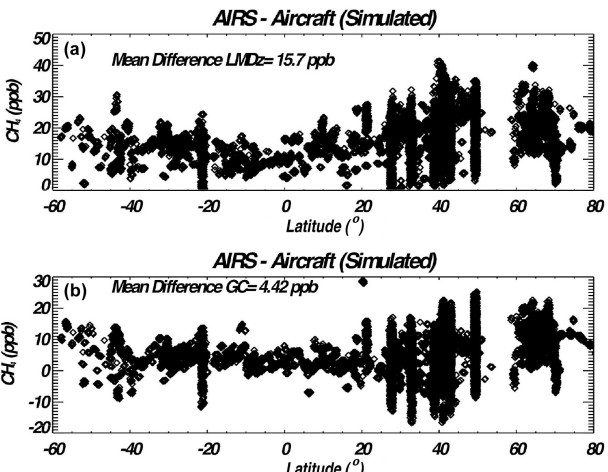

**Figure 4.** Simulated comparison between AIRS and aircraft in which the LMDz model **(a)** and GEOS-Chem model **(b)** are used for the simulation. This represents uncertainty in the true state that we validate against.

the Laboratoire de Météorologie Dynamique (LMDz) model (e.g., Folberth et al., 2006) model and the Goddard Earth Observing System (GEOS-Chem) model (e.g., Maasakkers et al., 2019). So, if the model value equaled the AIRS prior in the stratosphere, this difference would be zero. The differences for $x^c_{aircraft}$ from the LMDz model and GEOS-Chem are shown in Fig. 4 for all HIPPO ocean and land data; these differences show that model–model differences in the stratosphere can contribute significantly to the differences between AIRS and aircraft validation.

These differences provide an estimate for how knowledge error in the stratosphere projects to uncertainties in our methane retrievals. For example, this uncertainty varies with

latitude, similar to the residual bias between the AIRS estimate and aircraft (next section). Furthermore, the variability over small latitudinal ranges of 10° or less suggests that the random part of the stratospheric error is smaller than this latitudinal variability. Our estimate for this error is the average of these two errors, 10 ppb, and places an upper bound on the ability to validate AIRS CH$_4$. Our estimate for this error agrees with the 10 ppb estimate for the impact of stratospheric uncertainty on column estimates from aircraft profiles (Wunch et al., 2010). Appendix A shows further analysis of mean differences of AIRS minus aircraft for different profile extension choices. The bias varies by $\sim 5$ ppb for different profile extension choices when comparing at 700 hPa, $\sim 10$ ppb for different profile extension choices when comparing at 500 hPa, and $\sim 11$ ppb for different profile extension choices when comparing the column above 750 hPa.

The methane profile has a strong variable negative vertical gradient in the stratosphere. Models in general have a positive bias in the extratropical stratosphere (Patra et al., 2011). In GEOS-Chem $4 \times 5$, the column bias is shown in Fig. 2c of Turner et al. (2015) and further discussed in Maasakkers (2019), which resolves the bias to the stratosphere, and model stratospheric accuracy is an active research area (Ostler et al., 2016; Maasakkers et al., 2019).

### 3.4 Bias correction

AIRS CH$_4$ shows a persistent high bias of 25 to 90 ppb vs. aircraft observations in Fig. 5. Previous studies using remotely sensed measurements suggest that a bias correction to the AIRS methane profile measurement must account for the vertical sensitivity (e.g., Worden et al., 2011). For example, in the limit where the AIRS measurement is perfectly sensitive to the vertical distribution of methane, the bias correction could be a simple scaling factor. However, in the limit where the AIRS measurement is completely insensitive (e.g., DOFs = 0.0), then the bias correction is zero. We therefore use the bias correction approach described in Worden et al. (2011), where a bias profile (which varies by pressure) is passed through the averaging kernel to account for the AIRS sensitivity, as seen in Eq. (8). The form of the bias profile, $\delta_{bias}$, is set in Eq. (9).

We use HIPPO-4 observations to set a bias correction which we then evaluate with the other HIPPO campaigns and NOAA aircraft network data. HIPPO-4 was selected as it covers a wide range of latitudes and so that the bias correction can be set and tested with two independent datasets. To set the bias, we use Eq. (6b) to estimate the aircraft observation as seen by AIRS then compare this to AIRS observations. The result (by pressure level) is shown in Table 1. Then a bias was applied to AIRS using Eq. (8), with the bias term $\delta_{bias}$ in the form of Eq. (9).

$$\hat{x}_{corrected} = \hat{x}_{orig} + \mathbf{A}\delta_{bias}, \tag{8}$$

**Table 1.** Bias vs. pressure with and without bias correction. The bias correction was developed on HIPPO-4 and tested on HIPPO-4; HIPPO-1, HIPPO-2, HIPPO-3, and HIPPO-5; and the NOAA aircraft network.

| Pressure (hPa) | AIRS minus aircraft_AK (HIPPO-4) (ppb) | After bias correction (HIPPO-4) (ppb) | After bias correction (all HIPPO except HIPPO-4) (ppb) | After bias correction (all NOAA) (ppb) |
|---|---|---|---|---|
| 1000 | 24 | −1 | −3 | 1 |
| 824 | 36 | 0 | −4 | 1 |
| 681 | 48 | 1 | −5 | 2 |
| 562 | 58 | 1 | −4 | 2 |
| 464 | 60 | −5 | −3 | 3 |
| 383 | 67 | −5 | −2 | 2 |
| 316 | 81 | 1 | 4 | – |
| 261 | 86 | 1 | 4 | – |
| 215 | 89 | 1 | 3 | – |
| 161 | – | – | 4 | – |

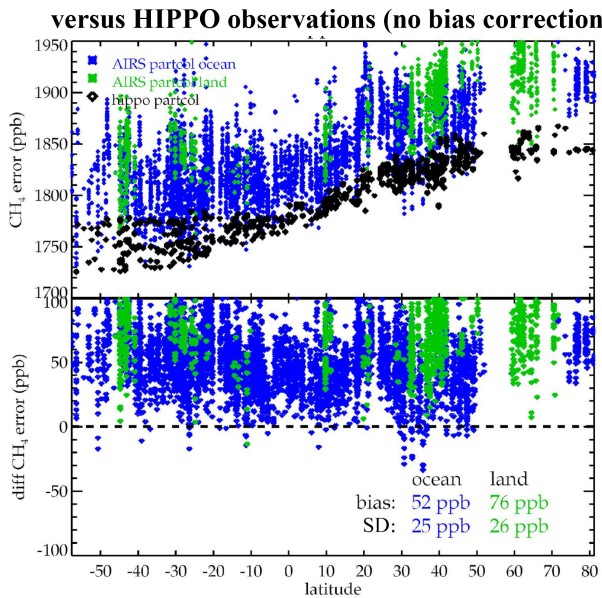

**Figure 5.** Comparison of AIRS methane VMR to aircraft for all HIPPO comparisons over the partial column XCH$_4$ VMR within the pressure levels measured by the aircraft. Blue shows AIRS ocean observations, and green shows AIRS land observations.

where $\hat{x} = \ln(\mathrm{VMR})$ because the retrieved quantity is $\ln(\mathrm{VMR})$, $\delta_{\mathrm{bias}}$ is a vector, and **A** is the averaging kernel matrix for $\hat{x} = \ln(\mathrm{VMR})$. We fit a single bias function for all AIRS measurements by minimizing the difference between AIRS and HIPPO-4, with $\delta_{\mathrm{bias}}$ constrained to have a slope with pressure and two pressure domains. We specify that $\delta_{\mathrm{bias}}$ cannot jump more than 0.05 (5 %) between the two domains.

$$\delta_{\mathrm{bias}} = c + d\,P \quad (P > P_o)$$
$$\delta_{\mathrm{bias}} = e + f\,P \quad (P < P_o) \tag{9}$$

where $P$ is pressure in hPa. The optimized bias correction parameters were $c = 0.0$; $d = -6.1 \times 10^{-5}$; $P_0 = 400\,\mathrm{hPa}$; $e =$

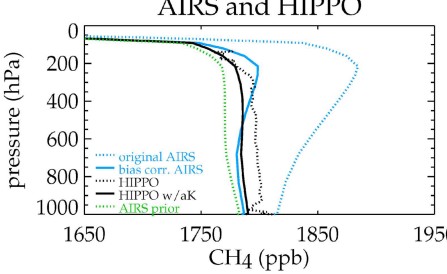

**Figure 6.** Example of the effect of bias correction on the AIRS profile from averaged HIPPO-1, HIPPO-2, HIPPO-3, and HIPPO-5. The blue lines show the AIRS methane profile before (dotted) and after (solid) bias correction. The black lines show the HIPPO measurements before (dotted) and after the averaging kernel is applied (solid).

$-0.09$; $f = 0.00018$. These bias correction results are shown for HIPPO-4; HIPPO-1, HIPPO-2, HIPPO-3, and HIPPO-5; and NOAA observations in Table 1. The remainder of the paper, unless specified, uses data bias-corrected by Eqs. (8) and (9).

Figure 6 shows the effect of bias correction on the average of all HIPPO (1, 2, 3, and 5) AIRS profiles. The bias correction improves the mean AIRS–aircraft difference and improves the pressure-dependent skew in the bias (Table 1). The HIPPO data are shown before and after the AIRS averaging kernel is applied (using Eq. 6b), which has the effect of bringing the HIPPO observations towards the AIRS prior. This is to match the imperfect sensitivity of satellite-based observations, which are similarly influenced by the prior.

## 4 Evaluation against aircraft data vs. latitude

### 4.1 Comparison of aircraft observations with and without bias correction

Figure 5 shows a comparison between all AIRS measurements within 50 km and 9 h of an aircraft measurement and the aircraft measurement. The quantity compared is the partial column XCH$_4$ VMR within the pressure levels measured from the aircraft. There is a mean bias of 57 ppb overall, $\sim$ 52 ppb for ocean, and $\sim$ 76 ppb for the land. The rms CE1 difference is $\sim$ 26 ppb. Furthermore, there appears to be latitudinal variations in the bias. For example, the mean difference between the AIRS and aircraft over the ocean for latitudes less than 20° S is $\sim$ 74 ppb, and for latitudes between 20° S and 20° N, this bias is $\sim$ 56 ppb.

Figure 7 shows the same comparisons as Fig. 5 after bias correction (described in Sect. 3.4). The mean bias is 1 ppb, and the rms difference is 24 ppb. The overall land bias is 12 ppb, and the overall ocean bias is −2 ppb. The bias calculated in Fig. 7 weights every point equally. Table A1 shows a slightly different result for these biases, where the bias is calculated by the campaign then averaged over all campaigns. In Table A1 the partial column XCH$_4$ VMR within the pressure levels measured by the aircraft has a bias of 16 ppb for land and −2 ppb for ocean. Note that the HIPPO land observations are primarily in Australia, New Zealand, and North America, whereas the ocean comparisons are in the mid-Pacific, as seen in Fig. 1. We expect the rms difference to be similar to the observation error, as the terms that make up the observation error are the primary source of variability in the observations (e.g., Worden et al., 2017b). The predicted observation error from Fig. 3 is 27 ppb and is consistent with the rms difference seen here, 23 ppb. However, knowledge of the stratosphere and validation uncertainty is potentially a large component of the latitudinal variability in the difference seen in the bottom panel of Fig. 7.

We also compare AIRS CH$_4$ observations to the NOAA aircraft network and ATom observations and find similar results as for HIPPO. Figure 8, discussed in Sect. 4.2, shows ATom results, and Fig. 9, discussed in Sect. 4.2, shows comparisons to a NOAA aircraft time series. The biases for different pressure ranges, campaigns, and surfaces are shown in Table A1. Table A3 shows the SD of AIRS minus validation by pressure and surface type, for single observations and daily and seasonal averages.

### 4.2 Errors in averaged AIRS data

Satellite data are typically averaged in order to improve the precision of a comparison between data and model. However, as shown in the previous figure, these data contain errors that vary with latitude. For example, knowledge error of the true profile in the stratosphere as well as errors in the jointly retrieved AIRS temperature and water vapor retrievals have

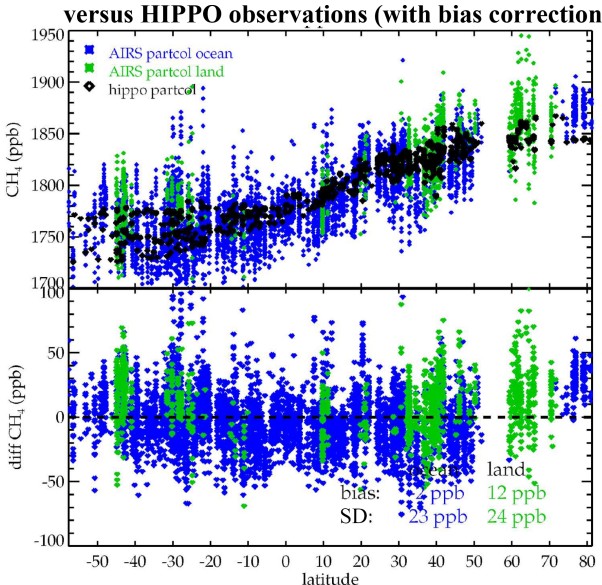

**Figure 7.** Same as Fig. 5 but after bias correction. The ocean has −2 ppb bias and 23 ppb SD, and the land has 12 ppb bias and 24 ppb SD.

both a random and a bias component, both of which vary with latitude. The bias component is approximately the same for all AIRS methane measurements taken on the same day within 50 km, as we do not expect large variations in temperature and water vapor errors over these scales, which we presume to be a driver of these correlated errors. To quantify the component of the accuracy that cannot be reduced by averaging, we compare averages of AIRS measurements to HIPPO and ATom measurements. We average over 1 d the AIRS observations matching a single HIPPO or ATom measurement, within ±50 km and 9 h of the measurement. We specify that there needs to be at least nine AIRS observations for each comparison so that the systematic error, and not the precision (or measurement error), is the dominant term. These daily AIRS averages contain, on average, 20 AIRS observations. Figure 8 shows the predicted error, assuming that the error is random, which is calculated by dividing the single observation error (24 ppb rms shown in Fig. 7) by the square root of the number of observations that are averaged. The mean predicted error for the averaged data, assuming random errors, is 6 ppb. The actual SD between the averaged AIRS and HIPPO or ATom data is $\sim$ 17 ppb, which is much larger and indicates that the errors within 1 d and 50 km are correlated. Note that the same-colored adjacent points (i.e., adjacent observations from the same campaign) often show similar biases. Because this rms difference is much larger than what would be expected if the errors were purely random, this shows the presence of systematic errors, either in the AIRS data or in the validation uncertainty. We therefore report 17 ppb as the limiting error when averaging AIRS data within 1° grids and 1 d

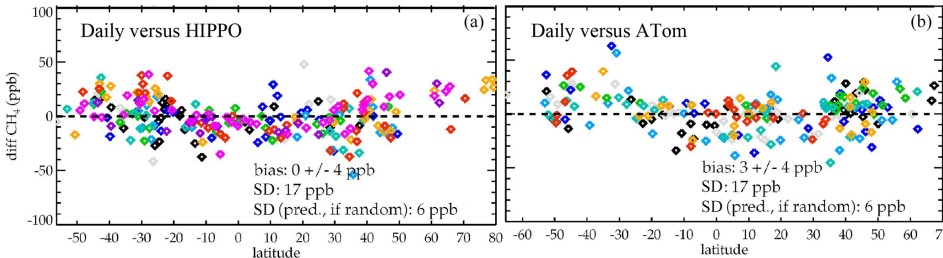

**Figure 8.** Comparison of daily averaged AIRS to HIPPO measurements **(a)** and ATom measurements **(b)** for the partial column observed by the aircraft. The different colors correspond to the campaigns shown in Fig. 1.

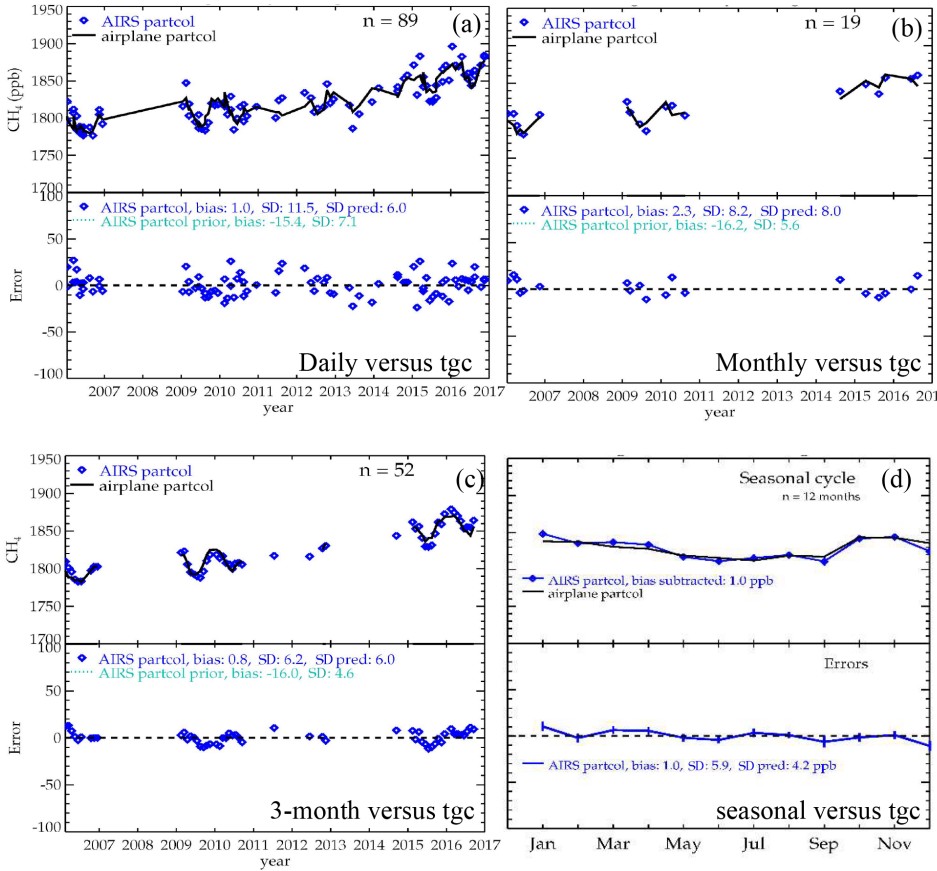

**Figure 9.** Comparison at TGC (27.7° N, 96.9° E). **(a, b)** Comparison of AIRS and co-located NOAA aircraft flights in SE Texas for the partial column XCH$_4$ VMR within the pressure levels measured by the aircraft. Data are averaged over **(a)** 1 d, **(b)** 1 month, and **(c)** 90 d, and averaged **(d)** by month from all years. **(c, d)** Difference from the aircraft. The predicted error for daily observations is the observation error (27 ppb) divided by the square root of the number of observations. The predicted monthly or seasonal error is the mean daily error (11.5 ppb) divided by the square root of the number of days averaged.

for the purpose of comparisons with models or other methane profiles.

On the other hand, averaging AIRS data seasonally can reduce the error further because geophysical errors such as temperature and water vapor vary over longer timescales. We demonstrate this aspect of the AIRS uncertainties by comparing averaged AIRS data to the NOAA aircraft methane profiles taken off the coast near Corpus Christi, Texas (27.7° N,

96.9° W, site TGC). We screen for at least three observations per day, fewer than the nine observations per day used for HIPPO and ATom daily averages in order to get enough daily averages to explore how the errors reduce with monthly and seasonal averages, since the aircraft make one–two measurements per month. Figure 9 shows daily, monthly, 90 d, and seasonal averages of the partial column XCH$_4$ VMR within the pressure levels measured from the aircraft at TGC. The

Atmos. Meas. Tech., 14, 1–20, 2021 https://doi.org/10.5194/amt-14-1-2021

seasonal averages are created by converting all AIRS–aircraft matched pairs to 2012 by adding 5.4 ppb yr$^{-1}$ multiplied by the year minus 2012 to account for the mean annual growth rate. The growth rate of 5.4 ppb yr$^{-1}$ is the mean increase during the AIRS record time period (2002–2019) estimated from the NOAA Global Monitoring Laboratory global surface measurements (https://esrl.noaa.gov/gmd/ccgg/trends_ch4/, last access: 21 December 2020). Since we are converting matched pairs of aircraft and AIRS to 2012, the differences between these matched pairs are unaffected by the accuracy of the conversion to 2012.

### 4.2.1 Daily average errors at TGC

We look at daily averages vs. aircraft data and find a similar result to that found with comparisons to ATom and HIPPO: daily averages have much larger errors than would be predicted if random errors are assumed. The SD of AIRS minus aircraft at TGC is 24 ppb, the SD for daily AIRS average minus aircraft is 11.5 ppb, as seen in Fig. 9a, and the predicted error for daily averages, assuming randomness in the error, is 6.0 ppb. Therefore, similarly to ATom and HIPPO, errors within 1 d and 50 km contain 11.5 ppb correlated error.

### 4.2.2 Monthly average errors at TGC

The NOAA aircraft measurements are usually taken about twice per month. The SD of monthly AIRS average minus aircraft is 8.2 ppb (Fig. 9b) for months containing more than one aircraft observation. This is compared to the daily error divided by the square root of the number of days averaged, 8.0 ppb. Therefore, errors for observations $\sim$ 2 weeks apart are uncorrelated.

### 4.2.3 3-month average errors at TGC

We average over 3-month scales, where averages must have at least 3 d. The SD of 3-month AIRS average minus aircraft is 6.2 ppb. The predicted error, taking the 11.5 ppb daily error and dividing it by the square root of the number of days averaged, is 6.0 ppb. Therefore, errors for 3-month averages are $\sim$ uncorrelated.

### 4.2.4 Seasonal cycle average errors at TGC

We average matched pairs within each month from any year. AIRS minus aircraft values for these averages have a SD of 5.9 ppb, whereas the predicted error, from the daily average divided by the square root of number of observations, is 4.2 ppb.

### 4.2.5 Summary of average errors at TGC

To summarize, averaging AIRS observations within 1 d reduces the error vs. aircraft, but correlated errors prevent daily averaged errors from dropping below 11.5 ppb. Averaging

daily averages over 1 or 3 months equals the daily error divided by the square root of the number of days averaged, indicating that errors are random in this domain. However, averaging months from multiple years does not reduce the error below 6 ppb, either due to correlated errors or validation uncertainty.

### 4.2.6 Summary of errors at all NOAA aircraft sites

Table A3 in Appendix A shows the single-observation SD for all NOAA aircraft sites. The ocean vs. land observations show similar values, with land and ocean SDs within 2 ppb. A single land observation has a SD vs. aircraft observations of 23 ppb for the partial column XCH$_4$ VMR within the pressure levels measured from the aircraft, in agreement with the predicted observation error of 23 ppb. The SD for daily averages is 15.2 ppb. This can be compared to the predicted error for the daily averages, assuming randomness, of 5.9 ppb. This indicates that there are correlated (non-random) errors on the order of 15 ppb when averaging observations within 50 km and 1 d. The monthly SD is 10.9, in reasonable agreement with the predicted of 9.4 ppb (from the daily average SD divided by the number of observations averaged). The seasonal cycle average, which is a monthly average of all matched pairs from all years, has a SD of 7.7 ppb, which is similar to the predicted error of 6.9 ppb (from the daily average divided by the square root of number of observations). We find that estimating the error as the daily SD divided by the square root of the number of days averaged is a reasonable estimate of the actual error.

### 4.2.7 The bias and bias uncertainty

The bias is estimated by calculating the mean bias for each campaign or station separately then calculating the mean and SD for all campaigns/stations. The bias vs. HIPPO is $0 \pm 4$ ppb. The bias vs. ATom is $3 \pm 4$ ppb. The bias vs. NOAA measurements is $9 \pm 7$ ppb.

## 5 Discussion and conclusions

We validate single-footprint AIRS methane by comparing 27 000 AIRS methane retrievals to 396 aircraft profiles from the HIPPO campaign, 719 profiles from the NOAA GML aircraft network, and 289 aircraft profiles from the ATom campaign, taken across a range of latitudes, longitudes, and times. The AIRS methane retrievals are derived using the MUSES optimal estimation algorithm that has previously been applied to Aura TES radiances (e.g., Fu et al., 2013). After adjusting the aircraft profile to account for the AIRS sensitivity (using the averaging kernel and a priori profile), we compare the mean methane value over the aircraft profile to the mean methane from the AIRS profile over the same altitude (or pressure) range. We use a subset of validation data to derive a pressure-dependent bias correction on the order

of $-60$ ppb and test this on an independent set of validation data. After the bias correction, we report a bias of $0 \pm 10$ ppb. The bias between AIRS and aircraft varies with pressure and location, as seen in Appendix A.

After applying the bias correction, from Eqs. (8) and (9), the rms difference between the AIRS and aircraft data of the partial column XCH$_4$ VMR within the pressure levels measured by the aircraft of $\sim 22$ ppb is consistent with the mean observation error, composed of random error from noise and the cross-state errors from jointly retrieved temperature, water vapor, clouds, and surface parameters that are projected onto the AIRS methane retrieval. The extent to which the aircraft profiles used here can be utilized as "truth" for the purposes of validation is limited by knowledge of the methane profile above the aircraft profile (referred to here as validation uncertainty), which limits our knowledge of "truth" to within about 10 ppb. This uncertainty is consistent with the location-dependent bias in the satellite–aircraft comparisons which can vary by $\sim 10$ ppb.

We quantify the AIRS minus validation SD for single observations, daily averages (within 50 km of the validation location), monthly averages, and seasonal averages for data bias-corrected using Eqs. (7) and (8). The AIRS minus validation SDs are 24 ppb (single AIRS footprint), 17 ppb (daily AIRS averages within 1degree latitude and longitude), 10 ppb ("monthly" AIRS averages), 9 ppb (3-month AIRS average), and 7 ppb (seasonal cycle average). The errors on averaged AIRS data are likely an upper bound on the AIRS error, due to the uncertainty in the validation. The single-footprint and daily average SDs for different pressure ranges and surface types are shown in Appendix A. We recommend using the SDs in this paragraph as the error budget for the specified averaged quantities.

These results can be compared to AIRS v6 validation by Xiong et al. (2015), which validated AIRS CH$_4$ retrieved from cloud-cleared radiances on the nine-footprint 45 km field of regard. Xiong et al. (2015) found AIRS SDs vs. HIPPO of 0.9 % (16 ppb) for pressures between 575 and 777 hPa, 1.2 % (18 ppb) SD for pressures between 441 and 575 hPa, and 1.6 % (29 ppb) between 343 and 441 hPa. Xiong et al. (2015) also found a pressure-dependent bias, with a $-25$ ppb bias near the top of the troposphere and a high 5 ppb bias near the mid-troposphere.

## Appendix A: Biases and SDs for different stations, campaigns, pressures, and surface types

We characterize the bias vs. validation data by station, campaign, and pressure level. Table A1 shows biases vs. validation data, after bias correction with Eqs. (8) and (9). In the HIPPO comparisons, the biases are generally smaller than about 10 ppb. There is no overall pattern in the bias by season. The land data are biased higher than ocean for HIPPO comparisons (about $+20$ ppb). However, note that the land observations vs. HIPPO are primarily in Australia and New Zealand, whereas the ocean comparisons are in the mid-Pacific.

The NOAA aircraft network comparisons are sorted by site. Many NOAA aircraft locations are at land–ocean interfaces, allowing a more direct comparison of the land–ocean biases. On average, the AIRS land observations are 0–5 ppb higher than AIRS ocean observations at the different pressures and pressure ranges. The overall bias of AIRS vs. NOAA aircraft is $+7.1$ ppb, whereas AIRS vs. HIPPO is 4.4 ppb for the partial column XCH$_4$ VMR within the pressure levels measured by the aircraft. This is consistent with AIRS land having a high bias vs. ocean of 0–5 ppb. The SD of the bias for the different campaigns is a useful quantity as it is an indication of systematic error. The SD of the bias varies from 4 to 9 ppb for the different vertical quantities.

Table A2 shows the mean bias for AIRS minus NOAA GML aircraft for land and ocean AIRS observations. The different rows extend the aircraft using the AIRS prior, the CarbonTracker model (from https://www.esrl.noaa.gov/gmd/ccgg/carbontracker-ch4/, last access: 21 December 2020) or the GEOS-Chem model (both are extended through 2018 using 2.5 % secular increase). The goal of this table is to approximate the influence of the profile extension on the validation accuracy.

Table A3 shows the SD for AIRS observations minus validation data for land–ocean for different pressure ranges for both single observations and AIRS averages. The mean bias at each site is subtracted prior to calculating the SD. This table shows the SDs for single observations and averaged quantities. The predicted error for the daily average is the observation error divided by the square root of the number of observations and is much smaller than the actual SD, indicating correlated errors. The predicted error for the monthly, 3-month, and seasonal cycle averages is the daily SD divided by the square root of the number of days averaged and $\sim$ agrees with the actual SD for the partial column XCH$_4$ VMR within the pressure levels measured by the aircraft. The location-dependent biases are subtracted from AIRS prior to calculating the SD in all but the last two rows. The last two rows show the SDs without subtracting the location-dependent biases, which increases the SD from about 8 to about 9 ppb.

**Table A1.** Bias by campaign, station, land–ocean, and pressure.

| Station/ campaign | Location | Time period | Bias 700 hPa (ppb) | Bias 500 hPa (ppb) | Bias 300 hPa (ppb) | Bias column matching aircraft (ppb) | Bias column above 750 hPa (ppb) |
|---|---|---|---|---|---|---|---|
| HIPPO 1S | Pacific | Jan, 2009 | −6.2 | 2.4 | 11.0 | 4.2 | 6.3 |
| HIPPO 1N | Pacific | Jan, 2009 | −3.2 | 3.7 | 12.5 | −0.1 | 4.8 |
| HIPPO 2S | Pacific | Nov, 2009 | −9.0 | −0.4 | 9.8 | −4.4 | 5.0 |
| HIPPO 2N | Pacific | Nov, 2009 | −4.3 | −3.3 | −3.1 | −4.0 | −4.0 |
| HIPPO 3N | Pacific | Apr, 2010 | −8.5 | 1.1 | 16.5 | −2.6 | 2.6 |
| HIPPO 4S | Pacific | Jun, 2011 | −0.7 | −2.0 | 9.5 | 1.8 | 10.2 |
| HIPPO 4N | Pacific | Jul, 2011 | 8.7 | 11.8 | 0.7 | 8.7 | 7.3 |
| HIPPO 5S | Pacific | Aug, 2011 | 1.2 | 7.6 | 13.3 | 4.5 | 9.3 |
| HIPPO 5N | Pacific | Sep, 2011 | −5.2 | 0.5 | 1.2 | −2.0 | 2.2 |
| HIPPO all land | – | – | 10.9 | 18.2 | 17.8 | 16.1 | 14.8 |
| HIPPO all ocean | – | – | −5.2 | −0.9 | 4.3 | −1.7 | 3.1 |
| HIPPO all (mean) | – | – | −2.9 | 2.1 | 7.9 | 0.7 | 4.9 |
| HIPPO all (SD) | – | – | 5.9 | 5.2 | 6.7 | 4.4 | 4.3 |
| ACG | 68° N, 152° W | – | 21.4 | – | – | 18.6 | 26.7 |
| ESP | 49° N, 126° W | – | 9.7 | – | – | 8.2 | 13.8 |
| NHA | 43° N, 71° W | – | 15.7 | 23.8 | – | 15.7 | 19.3 |
| THD | 41° N, 124° W | – | 13.6 | 21.7 | – | 14.0 | 21.2 |
| CMA | 39° N, 74° W | – | −0.2 | 5.7 | – | 0.9 | 3.6 |
| TGC | 28° N, 97° W | – | 1.0 | 7.9 | – | 2.3 | 6.5 |
| RTA | 21° S, 160° W | – | 3.7 | 11.5 | – | 3.9 | 12.8 |
| NOAA all land | – | – | 9.2 | 16.8 | – | 9.4 | 14.3 |
| NOAA all ocean | – | – | 9.0 | 12.8 | – | 8.7 | 15.4 |
| NOAA all (mean) | – | – | 9.3 | 14.1 | – | 9.1 | 14.8 |
| NOAA all (SD) | – | – | 8.1 | 8.2 | – | 7.1 | 8.2 |
| ATom 1S | Pacific | Aug, 2016 | −0.2 | 4.5 | 7.7 | 2.0 | 3.5 |
| ATom 1N | Atlantic | Aug, 2016 | 0.2 | 3.2 | 13.2 | 2.8 | 6.9 |
| ATom 2S | Pacific | Feb, 2017 | −6.8 | 0.7 | 8.4 | −2.5 | 5.2 |
| ATom 2N | Atlantic | Feb, 2017 | 5.7 | 12.3 | 25.3 | 8.3 | 12.5 |
| ATom 3S | Pacific | Oct, 2017 | −2.5 | 3.0 | 9.1 | 0.9 | 5.9 |
| ATom 3N | Atlantic/Pacific | Oct, 2017 | 6.5 | 13.0 | 21.9 | 9.3 | 13.8 |
| ATom 4S | Pacific | Apr/May, 2018 | −0.1 | 3.9 | 9.4 | 2.3 | 6.0 |
| ATom 4N | Atlantic | May, 2018 | −1.4 | 5.9 | 23.4 | 3.4 | 13.2 |
| ATom all land | – | – | 16.7 | 23.6 | 26.2 | 17.0 | 18.2 |
| ATom all ocean | – | – | −3.2 | 2.4 | 13.4 | 0.6 | 6.5 |
| ATom all (mean) | – | – | 0.1 | 5.8 | 14.7 | 3.2 | 8.3 |
| ATom all (SD) | – | – | 4.3 | 4.5 | 7.5 | 3.8 | 4.1 |

**Table A2.** Change in the mean bias of the partial column matching the NOAA aircraft observation using different aircraft profile extensions from the top aircraft measurement to the top of the atmosphere.

| Quantity | Profile extension | Bias 700 hPa (ppb) | Bias 500 hPa (ppb) | Bias 300 hPa (ppb) | Bias column matching aircraft (ppb) | Bias column above 750 hPa (ppb) |
|---|---|---|---|---|---|---|
| Land NOAA | CT | 6.0 | 10.3 | – | 6.1 | 3.8 |
| Ocean NOAA | CT | 4.5 | 5.7 | – | 4.3 | 4.0 |
| Land NOAA | prior | 9.2 | 16.8 | – | 9.4 | 14.3 |
| Ocean NOAA | prior | 9.0 | 12.8 | – | 8.7 | 15.4 |
| Land NOAA | GEOS-Chem | 6.4 | 11.7 | – | 6.7 | 6.4 |
| Ocean NOAA | GEOS-Chem | 4.4 | 7.7 | – | 4.5 | 6.4 |

**Table A3.** SD of AIRS minus validation for land–ocean observations and different pressures or pressure ranges. Rows 1–2 show the SD for single observation, rows 3–4 show the predicted observation error, rows 5–8 show the SD for daily averages, rows 9–10 show the predicted error for daily averages (assuming random error), rows 11–12 show the SD for 3-month averages, rows 13–14 show the SD for seasonal cycle averages (average the same month of all years), rows 15–16 show the predicted error for the seasonal cycle averages, and rows 17–18 show the SD without bias subtraction. The site-dependent biases from Table A1 are subtracted prior to calculating the SD.

| Quantity | SD 700 hPa (ppb) | SD 500 hPa (ppb) | SD 300 hPa (ppb) | SD column matching aircraft (ppb) | SD column above 750 hPa (ppb) |
|---|---|---|---|---|---|
| Land single | 26 | 29 | 26 | 23 | 25 |
| Ocean single | 25 | 27 | 26 | 22 | 24 |
| Land observation error | 26 | 26 | 19 | 23 | 19 |
| Ocean observation error | 28 | 28 | 20 | 24 | 19 |
| Land daily ($\geq 3$ obs d$^{-1}$) | 17 | 21 | 16 | 15 | 20 |
| Ocean daily ($\geq 3$ obs d$^{-1}$) | 18 | 21 | 21 | 16 | 20 |
| Land daily ($\geq 9$ obs d$^{-1}$) | 16 | 20 | 16 | 14 | 20 |
| Ocean daily ($\geq 9$ obs d$^{-1}$) | 17 | 19 | 21 | 15 | 18 |
| Land daily ($\geq 9$ obs d$^{-1}$) pred. | 9.7 | 9.9 | 5.7 | 8.5 | 7.0 |
| Ocean daily ($\geq 9$ obs d$^{-1}$) pred. | 8.4 | 7.9 | 4.6 | 7.0 | 5.7 |
| Land 3-month ($\geq 3$ obs d$^{-1}$, $\geq 3$ d) | 9.5 | 13.3 | – | 8.8 | 12.9 |
| Ocean 3-month ($\geq 3$ obs d$^{-1}$, $\geq 3$ d) | 9.0 | 11.8 | – | 8.3 | 11.8 |
| Land monthly (average all years) | 8.3 | 11.8 | – | 7.7 | 10.7 |
| Ocean monthly (average all years) | 8.3 | 10.4 | – | 7.5 | 10.1 |
| Land monthly (average all years) pred. | 7.7 | 9.9 | – | 6.9 | 9.3 |
| Ocean monthly (average all years) pred. | 8.0 | 9.8 | – | 7.2 | 9.5 |
| Land monthly (average all years) without bias subtraction | 9.9 | 13.7 | – | 9.1 | 12.2 |
| Ocean monthly (average all years) without bias subtraction | 10.4 | 12.3 | – | 9.4 | 11.6 |

*Data availability.* AIRS single-footprint methane data will be available at NASA GES DISC (https://disc.gsfc.nasa.gov/TS7) starting in January 2021. Note that the field "original_species" should be used with the bias correction described in this paper. The specific datasets used in this work are archived at https://drive.google.com/file/d/1crNs-QcOzbjiZUiTyRiTEsFORFTbODAW/view?usp=sharing (Kulawik et al., 2020). The NOAA GML aircraft observations were obtained from https://doi.org/10.25925/20190108TS8, obspack_ch4_1_GLOBALVIEWplus_v1.0_2019-01-08TS9. The ATom aircraft data were obtained from https://daac.ornl.gov/ATOM/TS10 (last access: February 2019).

*Supplement.* The supplement related to this article is available online at: https://doi.org/10.5194/amt-14-1-2021-supplement.

*Author contributions.* SSK and JRW are responsible for the study design, data analysis, and manuscript writing. VHP was responsible for data analysis and manuscript editing. DF was responsible for implementing AIRS into the MUSES retrieval system. SCW and BCD Jr. were responsible for HIPPO CH$_4$ data. KM and CS were responsible for the ATom CH$_4$ data. EJD, CS, and KM were responsible for NOAA GML aircraft data. AL, IP, YH, and KECP were responsible for implementation of the fast RTM, OSS, used in this work. YY provided LMDZ model runs. DJJ provided guidance on the GEOS-Chem model runs.

*Competing interests.* The authors declare that they have no conflict of interest.

*Acknowledgements.* This work is supported by NASA ROSES Aura Science Team NNN13D455T. Part of this research was carried out at the Jet Propulsion Laboratory, California Institute of Technology, under a contract with the National Aeronautics and Space Administration. The NOAA GML aircraft observations are supported by NOAA. The HIPPO aircraft data were supported by NOAA and NSF. Thanks are given to Bruce Daube, Eric Kort, Jasna Pittman, Greg Santoni and others for QCLS CH$_4$ data collection/processing. The GEOS-Chem model output is described in Worden et al. (2013a). Thanks are expressed for the helpful comments and feedback from Joannes D. Maasakkers.

*Financial support.* This research has been supported by the NASA (NASA ROSES Aura Science Team NNN13D455T).

*Review statement.* This paper was edited by Frank Keppler and reviewed by two anonymous referees.

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

**Remarks from the language copy-editor**

CE1     Please do give a short confirmation for each change. We normally ask again if there is no response to a comment, just to be sure that nothing is missed or misunderstood.

**Remarks from the typesetter**

TS1     Please check carefully that all changes have been inserted correctly.

TS2     Please confirm.

TS3     Please confirm running title.

TS4     Please carefully check that all vectors and matrices are denoted correctly. For future proofreadings, please highlight them in the PDF to avoid errors.

TS5     Please confirm.

TS6     According to our standards, changes like this must first be approved by the editor, as data have already been reviewed, discussed and approved. Please provide a detailed explanation for those changes that can be forwarded to the editor. Please note that this entire process will be available online after publication. Upon approval, we will make the appropriate changes. Thank you for your understanding.

TS7     Please provide date of last access.

TS8     Please provide full reference.

TS9     Is "obspack_ ch4_ 1_ GLOBALVIEWplus_ v1.0_ 2019-01-08" part of the DOI?

TS10     Please provide a reference list entry including creators and title.

TS11     Please provide place of publication.