# Peer review of "Evaluation of single-footprint AIRS CH4 Profile Retrieval"

_Atmospheric Measurement Techniques, 2020_

## Referee Comment (RC1) · Anonymous Referee #1 · 2 Jul 2020

It is challenging but worth to tackle to discuss how we perform meaningful comparisons between aircraft and satellite measurements that have different vertical resolution and altitude range and how we know errors inherent to satellite measurements. This paper tackles straightforwardly these issues and provides a useful perspective for validation work. The paper is suitable for the purpose of AMT journal and I recommend it to be published after some revisions.

Major comments:

1) "3. MUSES-AIRS Optimal Estimation of CH4 from single-footprint, original (non-cloud-cleared) AIRS radiances", page 5, "Good quality and sensitivity flagging for AIRS

[Figure]

CH4:" 1-1) The authors should give more explanations on the quality flags which they have applied to AIRS CH4 data. Are "radiance residual rms and mean" for all wavelength regions used in the AIRS CH4 retrieval? What are "NESR", "|KdotdL|", and "TSUR"? These terms may be referred to in some previous papers describing AIRS CH4 retrieval algorithm, but the authors should at least spell them out and give their definition. 1-2) "Cloud OD < 0.3" means that clouds that partially exist in the AIRS FOVs probably affect the AIRS retrievals and the amount of the effects would vary depending on the cloud OD itself. How did the authors evaluate the effects in this research? 1-3) How did the authors define tropopause height at each CH4 measurement location and calculate its tropospheric and stratospheric degrees of freedom? 1-4) What is "predicted error"? The authors discuss the "predicted error" in the later part of the text, but they should here mention the definition.

2) "3.1 Retrieval Error Characteristics", page 6, Equation (2) The authors should briefly explain the definition of A_xy. In this case, x indicates a methane profile and y does simultaneous retrieved parameters such as temperature and there is no relationship (cross-term) in nature between methane and temperature.

3) "3.2 Approach for Comparing AIRS measurements to aircraft profiles", page 7, Equation (4) In my understanding, A_cc is a symmetric matrix with a dimension of the number of atmospheric layers that can be observed by aircraft, while A matrix has a dimension of the number of full atmospheric layers from the surface to the top of the atmosphere. A_cs should be a non-square matrix, and how is it defined by the cross-terms of the A matrix? More explanation should be needed for readers.

4) "3.2 Approach for Comparing AIRS measurements to aircraft profiles", page 8, Equation (6) Sa matrix should in principle includes random errors only. The first term A_cb Sbb_a A_cb(T) comes from radiance biases that should have systematic characteristics. The other three terms or the second and the last terms only may be randomly distributed errors. Is it possible to treat errors that may have different characteristics in the same manner?

5) "3.2 Approach for Comparing AIRS measurements to aircraft profiles", page 9, lines 259-260 It should be more explained why "red" (mean obs. errors) minus "green" yields the cross-state error.

6) "3.3 Estimating validation error due to aircraft not measuring the stratosphere", page 9 I agree that the assumption in the stratosphere could significantly contribute to the differences between AIRS and aircraft data, but the amount of the validation error attributed to the stratosphere where aircraft cannot make observations should depend on how accurate each of the models (a priori, LMDz, and GC) can define the tropopause height; this may be a reason why we see relatively larger variabilities in the differences between AIRS and aircraft (simulated) in northern mid-latitudes.

7) Overall, it is better to describe a bit more clearly which of errors the authors think is a systematic or random one. The authors have already referred to which is which for each of the error components in several parts of the text, but there are too many error values and they sometimes resemble another one; it is an option to add a table to summarize the characteristics of each of the error components.

Minor comments:

1) Figure 1 It is easy to see if the international dateline is centered.

2) page 5, line 137 "... such as check on the " "2, residual signals, ...", a symbol before "2" is missing.

3) page 6, line 165 The symbols in the text do not correspond to those in the equations.

4) Figure 2 The authors should explain the color shading in more details; which pressure layers do colors indicate?

5) page 8, line 251 Which is correct, $A\_cn$ in the text or $A\_cs$ in Equation (6)? Or $A\_cs$ and $A\_cs$ are used as the same meaning?

6) Figure 3 The authors should give more explanations in the caption using the terms

in the equations in the text.

7) Figures 5 and 6 It may be better to replace the figure numbers.

8) page 10, lines 294-295 Why did the author choose HIPPO-4 observations to estimate bias correction values?

9) page 10, lines 295-296 Equation 5 is split into two: Eq. 5a and Eq. 5b. Are they combined?

10) Figures 6 and 7 The bias values shown in the figures do not correspond to the values in the text. The authors should explain the values in the captions of the figures.

11) page 12, line 350 Where does "24" come from?

12) page 12, line 363 It is better to add an explanation of 5.4 ppb (growth rate per year calculated at this site?).

13) page 12, lines 378-380 Does this sentence mean that there are some correlations among each of the differences between collocated AIRS and aircraft pairs and the correlations cannot be compensated when taking averages on a daily basis?

---

## Referee Comment (RC2) · Anonymous Referee #2 · 10 Jul 2020

General comment: The paper describes the validation of single footprint AIRS methane measurements by aircraft data from the HIPPO and ATom campaigns and NOAA aircraft measurements. The validation formalism is set up first, then the assessable components of the AIRS error budget as measurement error, validation error and bias are introduced and discussed. The paper closes with an assessment of the non-reducible error of AIRS methane averages.

Over all the paper is sound, the formalism is well introduced, and the error components are convincingly derived. I think the paper can be published after minor revisions, which are, however, numerous. I will list here the most important ones only. For the others I

attach the annotated manuscript.

Specific comments:

(1) During first reading I got confused at several instances because an error was introduced by number, while the path towards derivation of this error was only explained in the next para or section. I always thought I have missed a part of the manuscript but found out with re-reading that the explanation follows on the number. It would be easier (at least for readers like me) to first provide the explanation and then the number.

(2) The authors use very often the term "partial column". They need to introduce how the partial methane column over the pressure range the aircraft had measured has been calculated, both for the aircraft and AIRS data (what shape of the methane profile? where do pressure and temperature profiles come from?); further, the figures usually show volume mixing ratios (in ppb) instead of partial colums. This might sound picky, but I think this accurateness in wording should be kept.

(3) I was a bit surprised that the two models that were used to assess the so-called "validation error" (i.e. the unknown stratospheric part of the profile) provided largely different results (4.4 vs. 15.7 ppb). I think the authors should elaborate a little more on the reasons for this large difference. This is particularly important because the averaging kernels in Fig. 2 demonstrates that the stratospheric information is large mapped into the troposphere below 300 hPa, i.e. the AIRS signal obviously depends a lot on the assumptions about the stratosphere. Is it possible that the use of other models leads to even larger estimates of the validation error?

(4) A side remark without relevance to the revision of this paper: the SPARC TUNER activity has worked on recommendations about error reporting. A paper on this topic has just been accepted by AMT. It would be nice to look into that paper (amt-2019-350) and possibly following these recommendations in future.

Please also note the supplement to this comment:

https://www.atmos-meas-tech-discuss.net/amt-2020-145/amt-2020-145-RC2-supplement.pdf

**Supplement:**

[revised manuscript text omitted]

where $x\hat{}\_is$ the estimate of Log(VMR), $⟦x\_⟧\_a\hat{}$ is the log of the a priori concentration profile used to regularize the inversion. We split x into [x,y], where x is the quantity of interest, the methane profile, and y are the jointly estimated quantities (such as temperature, water vapor, clouds, and surface properties), which results in the cross-state error (Worden et al., 2004; Connor et al., 2008).

$$\hat{\mathbf{x}} = x_a + A_{xx}(x - x_a) + G_R \sum_i K_i^b (b_i - b_i^a) + A_{xy}(y - y_a) + Gn \tag{2}$$

For the AIRS (and TES) OE methane retrievals, $\mathbf{x}_a$ comes from the MOZART atmosphere chemistry model (e.g. Brasseur *et al.*, 1998). The vector $\mathbf{x}$ is the "true state", or in this case the (log) concentration profile. The matrix $\mathbf{A}$ is the averaging kernel matrix or $A = \frac{\partial \hat{x}}{\partial x}$ and describes the vertical sensitivity of the measurement. The matrix $\mathbf{G}$ relates changes in the radiance ($\mathbf{L}$) to perturbations in x, $G = \frac{\partial x \partial L}{\partial L \partial x}$. The vector $\mathbf{n}$ is the noise vector, the matrix $\mathbf{K}$ is the sensitivity of the radiance to changes in (log) concentration $K = \frac{\partial L}{\partial log(x)} = \frac{\partial L}{\partial log(VMR)}$, and the set of vectors $\mathbf{b}_i$ represent interference errors not estimated from the observed radiances. The true state, noise vector, and interference errors as described here are the "true" values and are therefore not actually known but are represented in this form so that we can calculate how their uncertainties affect the estimate $\hat{x}$. An example averaging kernel matrix is shown in Figure 2 and shows that AIRS based estimates of methane are most sensitive to methane in the free-troposphere and lower-stratosphere as demonstrated previously for AIRS and other TIR based estimates of tropospheric methane (e.g. Xiong *et al.*, 2016; de Lange and Landgraf, 2018).

Finally, we look at the quantity of interest, $\hat{x} = \mathbf{hx}$. The vector $\mathbf{h}$ combines all the necessary operations that maps the (log) concentration profiles to whatever quantity is needed such as selecting one particular pressure level (e.g. $\mathbf{h} = [0,0,0,1,0,0,0, …]$, selecting a column average, ($\mathbf{h}$ = pressure weighting function) – see Connor et al., 2008) or selecting the VMR mean (e.g. $\mathbf{h} = 1/m$, where m is the number of pressure levels to average).

$$\hat{x} = \mathbf{h}\hat{\mathbf{x}} \tag{3a}$$

$$\hat{x} = \mathbf{h}x_a + \mathbf{h}A_{xx}(x - x_a) + \mathbf{h}G_R \sum_i K_i^b (b_i - b_i^a) + \mathbf{h}A_{xy}(y - y_a) + \mathbf{h}
[revised manuscript text omitted]

730

---

## Author Comment (AC3) · 19 Sep 2020

Thank you to the reviewers for their helpful comments. The format of this response is alternating paragraphs of: reviewer comment, response, reviewer comment, response... The ordering of the responses are: major comments from reviewer 2 and minor comments from reviewer 2 (comments taken from the embedded PDF).

Reviewer 2 major comments:

Reviewer 2: (1) During first reading I got confused at several instances because an error was introduced by number, while the path towards derivation of this error was

only explained in the next para or section. I always thought I have missed a part of the manuscript but found out with re-reading that the explanation follows on the number. It would be easier (at least for readers like me) to first provide the explanation and then the number.

Response: A reference to the section where each is calculated was added, or the "preview" was removed. "Figure 3 shows the predicted errors for the AIRS partial column XCH4 VMR within the pressure levels measured by the aircraft. The measurement error (light green) is 18 ppb (from the last term of Eq. 7b) , and the total error for a single observation (including smoothing error) is 41 ppb. A component of the total error, the cross-state error, is estimated as 21 ppb (from Eq. 7b)." The smoothing error estimate now shown after the new Eq. 5 and the text states that this is calculated from Eq. 5.

Reviewer 2: (2) The authors use very often the term "partial column". They need to introduce how the partial methane column over the pressure range the aircraft had measured has been calculated, both for the aircraft and AIRS data (what shape of the methane profile? where do pressure and temperature profiles come from?); further, the figures usually show volume mixing ratios (in ppb) instead of partial colums. This might sound picky, but I think this accurateness in wording should be kept.

Response: At the beginning of Section 2, 2 sentences were added to describe how partial columns are created, "The retrieval estimates AIRS CH4 dry volume mixing ratio (VMR) profile. When a "partial column quantity" is validated, the retrieved CH4 profile is post-processed into partial column XCH4 VMR relative to dry air, with methodology from Connor et al. (2008) and Kulawik et al. (2017), where the VMR's at the pressure levels are weighted according to a pressure weighting function, resulting in a partial column volume mixing ratio (VMR).". Based on reviewer 2's comments here and elsewhere, the paper was previously not clear that the partial column quantity is XCH4 VMR. The wording was updated throughout the paper to indicate that the validated quantity was "the AIRS partial column XCH4 VMR within the pressure levels measured by the aircraft"

Reviewer 2 (3): I was a bit surprised that the two models that were used to assess the so-called "validation error" (i.e. the unknown stratospheric part of the profile) provided largely different results (4.4 vs. 15.7 ppb). I think the authors should elaborate a little more on the reasons for this large difference. This is particularly important because the averaging kernels in Fig. 2 demonstrates that the stratospheric information is large mapped into the troposphere below 300 hPa, i.e. the AIRS signal obviously depends a lot on the assumptions about the stratosphere. Is it possible that the use of other models leads to even larger estimates of the validation error?

Response: Added additional text and reference on the errors from model extension of the aircraft profile: "The methane profile has a strong, spatially varying negative vertical gradient in the stratosphere. Models in general have a positive bias in the extratropical stratosphere (Patra et al., 2011). In GEOS-Chem 4x5, the column bias is shown in Figure 2c of Turner et al. (2015) and further discussed in Maasakkers (2019), which resolves the bias to the stratosphere, and model stratospheric accuracy is an active research area (Ostler et al., 2016; Maasakkers et al., 2019)."

Reviewer 2 (4) A side remark without relevance to the revision of this paper: the SPARC TUNER activity has worked on recommendations about error reporting. A paper on this topic has just been accepted by AMT. It would be nice to look into that paper (amt-2019-350) and possibly following these recommendations in future.

Response: Thank you for this information It is good to standardize validation.

Embedded notes/comments from Referee 2. I summarized the comments and wrote the response.

1) Abstract: Define "validation error" Response: Update wording in abstract, and also change name to "validation uncertainty" "We estimate a 16 ppb validation uncertainty because the aircraft typically did not measure methane at altitudes where the AIRS measurements have some sensitivity, e.g. the stratosphere, and there is uncertainty in the truth that we validate against."

Section 2: Question about how the partial column is defined and the profile shape assumptions. The text was updated in response to this and to an earlier comment to clarify that a profile is retrieved, and then an XCH4 VMR is calculated. "The retrieval estimates AIRS CH4 dry volume mixing ratio (VMR) profile. When a "partial column quantity" is validated, the retrieved CH4 profile is post-processed into partial column XCH4 VMR relative to dry air, with methodology from Connor et al. (2008) and Kulawik et al. (2017), where the VMR's at the pressure levels are weighted according to a pressure weighting function, resulting in a partial column volume mixing ratio (VMR)."

3) cm-1 formatting. Response: Updated.

4) removed URL from citation as suggested by reviewer.

5) Request figure 1: center legend on Pacific. Update symbols. Remove gray color. Label aircraft sites. Response: Updated as rerquested

6) Description of Sa is needed. Response: Updated section around the new Equation 5, "The expected total error includes the smoothing error, which the covariance of the hcAccxc-x ac (Rodgers, 2000), where the covariance of xc-x ac is the a priori covariance, Saxx. The smoothing error is: (Eq 5) We estimate the smoothing error for the partial column XCH4 VMR within the pressure levels measured by the aircraft to be 30 ppb, using Eq. 5. This estimate strongly depends on Saxx, the a priori covariance, which is the same as in Worden et al. (2012); briefly 5% diagonal variability with correlations in pressure set from the MOZART model."

7) Description of the selection of quality flags. "The specific flags used for AIRS CH4 are as follows, which were set by minimizing the standard deviation of small clusters of retrievals and to standardize the sensitivity:"

8) The reviewer suggests that things like "|KdotdL| < 0.23 should be given proper names. Response: Since these variables are not referred to outside of this section, and additionally are referencing an existing data user's guide, the updates were to the
description of the equation, e.g. |KdotL| < 0.23 –> The absoluate value of KdotL < 0.23.

Equation 1, question about "G" versus "Gr". Response: Removed the R. Thanks.

Question about "sum over i" in Equation 1. Response: change b to a vector and the summation is implied just like all the other matrix multiplication.

Comment on A = dx/dx "Careful with log! Response: Yes, thank you, the equations are mixing up x and log(x). The intention was that x=log(VMR), so A = dx/dx, and K = dradiance/dx = dradiance/d(log(VMR)). Other issues fixed in this section from the reviewer.

Reviewer: recomment on estimate of smoothing error. Response: Addressed above, the smoothing error was estimated from the new Eq. 5.

Reviewer at Eq. 5b, "It is necessary to mention that this step is possible because the levels of the aircraft measurement are assumed uncorrelated." Response: This is now Eq. 6a, and the text was updated, "Equation 6a accounts for all of the AIRS smoothing error, whereas Equation 6b (the equation used in this work, other than Section 3.3) only accounts for the smoothing error from the part of the atmosphere measured by the aircraft profile. [...] The difference from Eqs. 6a and 6b is discussed in Section 3.3. "

Eq 6, "This step comes as surprise. Including 1-2 steps or explanation would be helpful." Response: Added new Eq. 5, introducing the a priori covariance, Eq. 7a, introducing the bias, and Eq. 7b, the covariance of 7a. Updated wording, "Equation 7a is the predicted bias between x_c (the measured AIRS value) and x_aircraft_c (the aircraft value with the AIRS Averaging kernel applied), and is the expected difference of Eqs. 4 and 6b. Equation 7b is the covariance of Eq. 7a, and estimates the predicted error."

Equation 6, "Acb is not defined". Response: change Acb to GcKb.

Line 235, "Here the explanation should be given that this is because the retrieval vector

is log(VMR)." Response: updated. "Because the retrieved quantity log(VMR), the error in ppb is approximately the fractional error times the methane value in ppb."

Line 255, "Define validation error". Response: Now it is consistently called "validation uncertainty" throughout the paper rather than "validation error" and "validation uncertainty".

Line 259, "Where do these errors come from?". Response: These errors are now tied to equations.

Line 266, The reviewer notes that not only the stratosphere influences the troposphere, but the troposphere influences the stratosphere, due to the broad sensitivity. Response: Added sentence, "Similarly, the true state in the troposphere influences retrieved values in the stratosphere."

Line 282. Is the 16 ppb the larger of the two errors from model propagation? Response: We updated this to set the "validation uncertainty" to the average of the two model results, 10 ppb. This agrees with the previous result from Wunch et al. (2010).

Reviewer comment: How does the profile extension relate to the "validation uncertainty"? Doesn't it relate more to the bias estimate? Response: The profile extension obviously affects the bias versus validation data which affects the bias correction. However, it is most relevant to the uncertainty that results from extending the aircraft profile using a model. Text update, "Appendix A shows further analysis of mean differences of AIRS minus aircraft for different profile extension choices. The bias varies by $\sim$ 5 ppb for different profile extension choices when comparing at 700 hPa, $\sim$10 ppb for different profile extension choices when comparing at 500 hPa, and $\sim$11 ppb for different profile extension choices when comparing the column above 750 hPa."

Line 291. Reviewer requests that Worden et al., 2011 bias correction be summarized briefly. Response: Added text, "We therefore use the bias correction approach described in Worden et al. (2011), where a bias profile (which varies by pressure) is

passed through the averaging kernel to account for the AIRS sensitivity, as seen in Eq. 8. The form of the bias profile, delta_bias set by Eq. 9."

Eq 7. Reviewer wonders if A is a vector. Response: A is a matrix, as x-hat is a profile, as shown in Fig. 6. Text updated, "Where x = ln(VMR), because the retrieved quantity is ln(VMR), delta_bias is a vector, and A is the averaging kernel matrix for x = ln(VMR)"

Eq 7. Reviewer wonders about the handling of log. Response: text and equations updated to handle log consistently.

Line 315. Reviewer wonders why there is a bias peak about 200 hPa. Response: It seems that the bias is increasing through the stratosphere, whereas the sensitivity is decreasing above 100 hPa, and these two combine to generate a peak about 200 hPa. But this is speculation.

Section 4 title changed from "Evaluation against aircraft data by latitude" to "Evaluation against aircraft data versus latitude", because as the reviewer points out, there is no fitting of bias or skill as a function of latitude.

Line 319 Reviewer suggests wording update to, "Figure 6 shows a comparison between all AIRS measurements within 50 km and 9h of an aircraft measurement over the pressure range of the partial column measured by the aircraft.". Response: The wording was updated to, "Figure 5 [figure numbering update] shows a comparison between all AIRS measurements within 50 km and 9h of an aircraft measurement and the aircraft. The quantity compared is the partial column XCH4 VMR within the pressure levels measured by the aircraft."

Figure 7 legend unreadable. Response: Add text in the figure caption with the information in the legend.

Line 334-335, reviewer questions what the legend means, e.g. 0.0 +/- 4. Response: This was not well explained in the text previously. This is an estimate of the overall bias and uncertainty in the bias. Each campaign or station's bias is independently calculated

and then the mean and standard deviation are calculated. A new sub-section was added at the end of Section 4.1, "4.2.6 The bias and bias uncertainty The bias is estimated by calculating the mean bias for each campaign or station separately, then calculating the mean and standard deviation for all campaigns / stations. The bias versus HIPPO is 0 ± 4 ppb. The bias versus ATom is 3 ± 4 ppb. The bias versus NOAA measurements is 9 ± 7 ppb."

Reviewer wants "roughly the same location and time" defined in line 344. Response: Text added to clarify, "The bias component is approximately the same for all AIRS methane measurements taken on the same day within 50 km, as we do not expect large variations in temperature and water vapor errors over these scales, which we presume to be a driver of these correlated errors."

Reviewer says this sentence is hard to parse, "The number of AIRS observations averaged ranges from 9 to 53 and averages 20.". Response: This was updated to, "We average over 1 day, the AIRS observations matching a single HIPPO or ATom measurement, within +-50 km and 9 hours of the measurement. We specify that there needs to be at least 9 AIRS observations for each comparison so that the systematic error, and not the precision (or measurement error), is the dominant term. These daily AIRS averages contain, on average, 20 AIRS observations."

In Figure 9, the reviewer noted that the figure showed VMR but was indicated as a partial column. Response: This is an XCH4 partial column, as noted above and now explained in the paper.' The wording describing the partial column throughout the paper, to remove this confusion, is, "The partial column XCH4 VMR within the pressure levels measured by the aircraft".

365-375. The reviewer says that this section is hard to follow and needs explanation on the importance of the values shown. Response: Updated this text to indicate what the numbers mean, and what the findings are. Also separate into 4 new subsections, discussing daily, monthly, 3-month, and seasonal cycle averaging. Added a new section

on the bias characterization (in response to a previous comment.

The rewritten text is (the next 7 paragraphs):

"4.2.1 Daily average errors at TGC We look at daily averages versus aircraft data, and find a similar result as found with comparisons to Atom and HIPPO: daily averages have much larger errors than would be predicted if random errors are assumed. The standard deviation of (AIRS minus aircraft) at TGC is 24 ppb, the standard deviation for (daily AIRS average minus aircraft) is 11.5 ppb, as seen in Fig. 9a, and the predicted error for daily averages, assuming randomness in the error, is 6.0 ppb. Therefore, similarly to ATom and HIPPO, errors within 1 day and 50 km contain 11.5 ppb correlated error.

"4.2.2 Monthly average errors at TGC The aircraft measurements are usually taken about twice per month. The standard deviation of (monthly AIRS average minus aircraft) is 8.2 ppb (Figure 9b) for months containing more than 1 aircraft observation. This is compared to the daily error divided by the square root of the number of days averaged, 8.0 ppb. Therefore, errors for observations ∼2 weeks apart are uncorrelated.

"4.2.3 3-month average errors at TGC We average over 3-month scales, where averages must have at least 3 days. The standard deviation of (3-month AIRS average minus aircraft) is 6.2 ppb. The predicted error, taking the 11.5 ppb daily error and dividing by the square root of the number of days averaged, is 6.0 ppb. Therefore, errors for 3-month averages are ∼uncorrelated.

"4.2.3 Seasonal cycle average errors at TGC We average matched pairs within each month from any year. (AIRS minus aircraft) for these averages, have a standard deviation of 5.9 ppb, whereas the predicted error, from the daily average divided by the square root of number of observations, is 4.2 ppb.

"4.2.4 Summary of average errors at TGC To summarize, averaging AIRS observations within one day reduces the error versus aircraft, but correlated errors prevent daily averaged errors from dropping below 11.5 ppb. Averaging daily averages over 1 or 3 months equals the daily error divided by the square root of the number of days averaged, indicating that errors are random in this domain. However, averaging months from multiple years, does not reduce the error below 6 ppb, either due to correlated errors, or validation uncertainty.

"4.2.5 Summary of errors at all NOAA aircraft sites Table A.3 in Appendix A shows the single-observation standard deviation for all NOAA aircraft sites. The ocean vs. land observations show similar values, with land and ocean standard deviations within 2 ppb. A single land observation has a standard deviation versus aircraft observations of 23 ppb for the partial column XCH4 VMR within the pressure levels measured from the aircraft, in agreement with predicted observation error of 23 ppb. The standard deviation for daily averages is 15.2 ppb. This can be compared to the predicted error for the daily averages, assuming randomness, of 5.9 ppb. This indicates that there are correlated (non-random) errors on the order of 15 ppb when averaging observations within 50 km and 1 day. The monthly standard deviation is 10.9, in reasonable agreement with the predicted of 9.4 ppb (from the daily average standard deviation divided by the number of observations averaged). The seasonal cycle average, which is a monthly average of all matched pairs from all years, has a standard deviation of 7.7 ppb, which is similar to the predicted error of 6.9 ppb (from the daily average divided by the square root of number of observations). We find that estimating the error as the daily standard deviation divided by the square root of the number of days averaged is a reasonable estimate of the actual error.

"4.2.6 The bias and bias uncertainty The bias is estimated by calculating the mean bias for each campaign or station separately, then calculating the mean and standard deviation for all campaigns / stations. The bias versus HIPPO is $0 \pm 4$ ppb. The bias versus ATom is $3 \pm 4$ ppb. The bias versus NOAA measurements is $9 \pm 7$ ppb."

Other changes:

Author institution updated for Edward J. Dlugokencky to "National Oceanic and Atmospheric Administration, Global Monitoring Laboratory, Boulder, CO, USA"

NOAA ESRL aircraft changed to "NOAA GML aircraft network" due to an updated name for this program.

Thank you to both reviewers for their helpful comments. We have responded to all reviewer comments.

---

## Author Comment (AC1)

**Supplement 1. Daily, monthly, 3-month, and seasonal plots at all NOAA ESRL sites between 2006 and 2017.**

These supplementary plots show daily, monthly, 3-month, and seasonal plots for the 7 aircraft sites used for timeseries comparisons to AIRS. The vertical quantities plotted are: the partial column XCH4 VMR within the pressure levels measured by the aircraft and the aircraft with the AIRS averaging kernel applied (Eq. 6b). Note that most sites take data once or twice a month. Note that other than 2006, our AIRS dataset has gaps (e.g. 2008) causing gaps in these timeseries. Monthly averages have a cutoff of at least 2 observation and 3-month averages have a cutoff of at least 3 observations to test the effect of averaging. The "std pred" is the daily standard deviation divided by the square root of the average number of observations averaged. The seasonal plots convert matched pairs of AIRS + aircraft to 2012 by adding 5.4 ppb per year multiplied by (year minus 2012), then averaging all values in a month. This is done to see if there is a persistent seasonal issue. The mean bias at each site is separately subtracted from the seasonal cycle as indicated on the plot.

The sites are listed here.

| | | |
|------|-----|------|
| ACG | 68N | 152W |
| ESP | 49N | 126W |
| NHA | 43N | 71W |
| THD | 41N | 124W |
| CMA | 39N | 74W |
| TGC | 28N | 97W |
| RTA | 21S | 160W |

ACG, Alaska Coast Guard, United States (68N, 152W)

Since 1-month averages have a cutoff of 2, and 3-month averages have a cutoff of 3, there are very few values.

[Figure]

ESP, Estevan Point, British Columbia (49N, 126W)

[Figure]

**NHA, Offshore Portsmouth, New Hampshire (Isles of Shoals) (43N, 71W)**

[Figure]

**THD, Trinidad Head, California (41N, 124W)**

[Figure]

CMA, Offshore Cape May, New Jersey (39N, 74W)

[Figure]

**TGC Offshore Corpus Christi, Texas (28N, 97W)**

[Figure]

**RTA, Rarotonga (21S, 160W)**

[Figure]

**Supplement 2.  Bias versus degrees of freedom, cloud optical depth, and pointing angle.**

This supplementary plot shows the AIRS the partial column XCH4 VMR above 750 hPa and the aircraft with the AIRS averaging kernel applied (Eq. 6b) for all HIPPO campaigns.  The y-axis shows the difference of AIRS and the aircraft, and the x-axis are:  AIRS degrees of freedom, AIRS cloud optical depth, and AIRS pointing angle.

---

## Author Response (AR1)

Thank you to both reviewers for their helpful comments. We have responded to all reviewer comments.

Other changes in addition to changes from the responses to reviewers:

Author institution updated for Edward J. Dlugokencky to "National Oceanic and Atmospheric Administration, Global Monitoring Laboratory, Boulder, CO, USA"

NOAA ESRL aircraft changed to "NOAA GML aircraft network" due to an updated name for this program.

The format of this response is alternating paragraphs of: reviewer comment, response, reviewer comment, response...

Response to reviewer 1. The ordering of the responses are: major comments from reviewer 1, minor comments from reviewer 1.

Responses to Major comments from Reviewer 1:

Reviewer 1: 1-1) 3. MUSES-AIRS Optimal Estimation of CH4 from single-footprint, original (non cloud- cleared) AIRS radiances:, page 5, "Good quality and sensitivity flagging for AIRS CH4:" 1-1) The authors should give more explanations on the quality flags which they have applied to AIRS CH4 data. Are "radiance residual rms and mean" for all wavelength regions used in the AIRS CH4 retrieval? What are "NESR", "|KdotdL|", and "TSUR"? These terms may be referred to in some previous papers describing AIRS CH4 retrieval algorithm, but the authors should at least spell them out and give their definition.

Response: A description of the quality flags was updated and a references given that defines and describes each quality flag. Text added to the paper, "Quality flags are discussed in more detail in the Aura-TES user's guide (pp 27-30, Herman et al., 2018). The specific flags used for AIRS CH4 are as follows, which were set by minimizing the standard deviation of small clusters of retrievals and to standardize the sensitivity:"

Reviewer 1: 1-2) "Cloud OD < 0.3" means that clouds that partially exist in the AIRS FOVs probably affect the AIRS retrievals and the amount of the effects would vary depending on the cloud OD itself. How did the authors evaluate the effects in this research?

Response: A plot of error versus cloud optical depth was added to the supplement. Text added, "* Cloud optical depth < 0.3. This ensures that the cloud is not opaque and there is fairly uniform sensitivity so that the bias correction is fairly consistent. The bias versus cloud optical depth is shown in the supplement."

Reviewer 1: 1-3) How did the authors define tropopause height at each CH4 measurement location and calculate its tropospheric and stratospheric degrees of freedom? 1-4)

Response: The tropopause height was obtained from GMAO files. Text added to the paper after Eq. 2: "The degrees of freedom, DOFs, describing the sensitivity of x to the true state, and is equal to the trace of A_xx. The degrees of freedom in the troposphere is equal to the trace of the averaging kernel corresponding to the troposphere, and the degrees of freedom in the stratosphere is equal to the trace of the averaging kernel corresponding to the stratosphere. The troposphere is defined using the tropopause height parameter from version 5 of the NASA Global Modeling and Assimilation Office (GMAO) Goddard Earth Observing System (GEOS-5) model (Molad et al., 2012)."

Reviewer 1: 1-4) What is "predicted error"? The authors discuss the "predicted error" in the later part of the text, but they should here mention the definition.

Response: The predicted error is the total error from the linear estimate, Eq. 4, and is a field in the output product. Text added to the paper after Eq. 7 "The square root of 7b is the predicted observation error.", and text added to the description of Fig. 3, "Figure 3 shows the predicted errors for the AIRS partial column XCH4 VMR within the pressure levels measured by the aircraft. The measurement error (light green) is 18 ppb (from the last term of Eq. 7b) , and the total error for a single observation (including smoothing error) is 41 ppb. A component of the total error, the cross-state error, is estimated as 21 ppb (from Eq. 7b)."

Reviewer 1: 2) 3.1 Retrieval Error Characteristics, page 6, Equation (2) The authors should briefly explain the definition of $A\_{xy}$. In this case, x indicates a methane profile and y does simultaneous retrieved parameters such as temperature and there is no relationship (cross-term) in nature between methane and temperature.

Responses: Added in new text after Eq. 2 "Axx describes the dependence of x on the true state x, and Axy describes the dependence of x on the true state y, which is non-zero because of correlations in the Jacobians, K, for x and y."

Reviewer 1: 3) "3.2 Approach for Comparing AIRS measurements to aircraft profiles", page 7, Equation (4) In my understanding, A_cc is a symmetric matrix with a dimension of the number of atmospheric layers that can be observed by aircraft, while A matrix has a dimension of the number of full atmospheric layers from the surface to the top of the atmosphere. A_cs should be a non-square matrix, and how is it defined by the cross-terms of the A matrix? More explanation should be needed for readers.

Response: Added in new text after Eq. 4 "So, if for example, the aircraft measured pressures 0-9, and did not measure pressure levels 10-65, then Acc=A[0:9,0:9] and Acs=A[0:9,10:65], where A is the full averaging kernel."

Reviewer 1 4) "3.2 Approach for Comparing AIRS measurements to aircraft profiles", page 8, Equation (6) Sa matrix should in principle includes random errors only. The first term A_cb Sbb_a A_cb(T) comes from radiance biases that should have systematic characteristics. The other three terms or the second and the last terms only may be randomly distributed errors. Is it possible to treat errors that may have different characteristics in the same manner?

Response: Equation 6, as the reviewer notes, only addressed random errors. Added Equation 6a, which is the bias component of error, and Equation 6b, the current Equation 6, the variable component of the error. Added additional text to describe this. "Equation 6a represents the propagation of mean biases from: (1) fixed (non-retrieved) parameters, e.g. spectroscopy (b), (2) jointly retrieved parameters, e.g. temperature, (y), (3) "stratospheric", describing the impact of the part of the atmosphere not covered by the aircraft on the measured part (xs), or (4) measurement errors (n) into biases of xc. The mean bias from 6a is difficult to characterize theoretically and is characterized during validation, and assumed to be primarily from the first term (e.g. spectroscopy). Equation 6b represents the propagation of these same error types into a varying error. Although Eq. 6b has overall zero bias, it can produce regional and temporal biases, e.g. as seen in Connor et al. (2016), where these biases approach zero over long enough spatial or temporal scales. The error covariances all represent fractional errors, in log(VMR). The error in ppb is approximately the fractional error times the methane value in ppb."

Reviewer 1, 5): "3.2 Approach for Comparing AIRS measurements to aircraft profiles", page 9, lines 259-260 It should be more explained why "red" (mean obs. errors) minus "green" yields the cross-state error.

Response: We removed the statement in question. It now reads, "Figure 3 shows the predicted errors for the AIRS partial column XCH4 VMR within the pressure levels measured by the aircraft. The measurement error (light green) is 18 ppb (from the last term of Eq. 7b) , and the total error for a single observation (including smoothing error) is 41 ppb. A component of the total error, the cross-state error, is estimated as 21 ppb (from Eq. 7b)."

Reviewer 1 6) "3.3 Estimating validation error due to aircraft not measuring the stratosphere", page 9 I agree that the assumption in the stratosphere could significantly contribute to the differences between AIRS and aircraft data, but the amount of the validation error attributed to the stratosphere where aircraft cannot make observations should depend on how

accurate each of the models (a priori, LMDz, and GC) can define the tropopause height; this may be a reason why we see relatively larger variabilities in the differences between AIRS and aircraft (simulated) in northern mid-latitudes."

Response: We agree that the stratospheric error depends on the accuracy of the model used to extend the validation data. We estimated the error for the model that we used in our validation. Added statement on page 9 "This estimate depends on the accuracy of the model used to extend the aircraft profile during the validation process, and was estimated for the model that we used in validation."

Reviewer 1 7) Overall, it is better to describe a bit more clearly which of errors the authors think is a systematic or random one. The authors have already referred to which is which for each of the error components in several parts of the text, but there are too many error values and they sometimes resemble another one; it is an option to add a table to summarize the characteristics of each of the error components.

Response: Added statement after Eq. 6b to indicate that the overall bias primarly results from the first term in Eq. 6a. However, the split of errors into "random" and "systematic" is not straightforward. Added a statement after Eq. 6b, "Although Eq. 6b has overall zero bias, it can produce regional and temporal biases, e.g. as seen in Connor et al. (2016), where these biases approach zero over long enough spatial or temporal scales."

Minor comments from reviewer 1.

1) Figure 1 It is easy to see if the international dateline is centered.

Response: Referee suggests switching view on Figure 1. This was updated, as well as the colors and symbols used, as requested elsewhere.

2) page 5, line 137 "... such as check on the " "2, residual signals, ...", a symbol before "2" is missing.

Response: Fixed. "We use similar quality flags as the TES retrievals such as checks on the radiance residual, ..."

3) page 6, line 165 The symbols in the text do not correspond to those in the equations.

Response: Fixed.

4) Figure 2 The authors should explain the color shading in more details; which pressure layers do colors indicate?

Response: The levels are listed in the figure caption with several of the pressures shown on the plot itself.

5) page 8, line 251 Which is correct, $A_{cn}$ in the text or $A_{cs}$ in Equation (6)? Or $A_{cs}$ and $A_{cs}$ are used as the same meaning?

Response: The reviewer helpfully pointed out inconsistent notation. We did a switch from calling the non-measured part of the atmosphere "n" to "s" during the paper formation, so all the Acn should be Acs. Updated the text. Thank you.

6) Figure 3 The authors should give more explanations in the caption using the terms in the equations in the text.

Response: We now show the equation for the "smoothing error" in Eq. 5. Label Eq. 7 as the "observation error". The text for Figure 3 now reads, "The total error shown is the smoothing error (Eq. 5) plus the observation error (Eq. 7b). The measurement error is the last term of Eq. 7b, and the only fully random error."

7) "Figures 5 and 6 it may be better to replace the figure numbers".

Response: I switched these two figures (was a clearer note in embedded comments in paper)

8) page 10, lines 294-295 Why did the author choose HIPPO-4 observations to estimate bias correction values?

Response: Added a sentence to describe why HIPPO-4 was chosen, "HIPPO-4 was selected as it covers a wide range of latitudes and so that the bias correction can be set and tested with two independent datasets."

9) page 10, lines 295-296 Equation 5 is split into two: Eq. 5a and Eq. 5b. Are they combined?

Response: Yes, Eq. 5 was split into 5a and 5b, but later referred to as "Equation 5". The text was updated here and one other place to refer to Eq. 5b.

10) Figures 6 and 7 The bias values shown in the figures do not correspond to the values in the text. The authors should explain the values in the captions of the figures.

Response: Thank you for noticing that. New text is added for Fig. 5 and 7 (Fig 5 & 6 were swapped based on the previous reviewer comments). The values in the paper were outdated, and updated. Here is the updated text describing the biases shown in Fig 7, "Figure 7 shows the same comparisons as Fig. 5 after bias correction (described in Section 3.4). The mean bias is 1 ppb, and the RMS difference is 24 ppb. The overall land bias is 12 ppb, and the overall ocean bias is -2 ppb. The bias calculated in Fig. 7 is weighs every point equally. Table A.1 shows a slightly different result for these biases, where the bias is calculated by campaign, then averaged over all campaigns. In Table A.1 the partial column XCH4 VMR within the pressure levels measured by the aircraft has a bias of 16 ppb for land, and -2 ppb for ocean."

11) page 12, line 350 Where does "24" come from?

Response: This comes from the single observation RMS, shown in Fig. 7. The text is updated to "Figure 8 shows the predicted error, assuming that the error is random, which is calculated by dividing the single observation error (24 ppb RMS shown in Fig. 7) by the square root of the number of observations that are averaged. The mean predicted error for the averaged data, assuming random errors, is 6 ppb. The actual standard deviation between the averaged AIRS and HIPPO or ATom data is ~17 ppb, which is much larger and indicates that the errors within 1 day and 50 km are correlated."

12) page 12, line 363 It is better to add an explanation of 5.4 ppb (growth rate per year calculated at this site?).

Response: Added text "The growth rate of 5.4 ppb/year is the mean increase during the AIRS record time period (2002-2019) estimated from the NOAA Global Monitoring Laboratory global surface measurements (https://esrl.noaa.gov/gmd/ccgg/trends_ch4/). Since we are converting matched pairs of aircraft and AIRS to 2012, the differences between these matched pairs is unaffected by the accuracy of the conversion to 2012."

13) page 12, lines 378-380 Does this sentence mean that there are some correlations among each of the differences between collocated AIRS and aircraft pairs and the correlations cannot be compensated when taking averages on a daily basis?

Response: Yes. Updated the wording to say this more clearly, "The standard deviation for daily observations is 15.2 ppb. This can be compared to the predicted error assuming randomness of 5.9 ppb (23 ppb divided by the square root of the number of observations averaged) Since 15.2 ppb is much larger than 5.9 ppb, this indicates that there are correlated (non-random) errors on the order of 15 ppb when averaging nearby observations within 1 day."

The format of this response is alternating paragraphs of: reviewer comment, response, reviewer comment, response... The ordering of the responses are: major comments from reviewer 2 and minor comments from reviewer 2 (comments taken from the embedded PDF).

Reviewer 2 major comments:

Reviewer 2: (1) During first reading I got confused at several instances because an error was introduced by number, while the path towards derivation of this error was only explained in the next para or section. I always thought I have missed a part of the manuscript but found out with re-reading that the explanation follows on the number. It would be easier (at least for readers like me) to first provide the explanation and then the number.

Response: A reference to the section where each is calculated was added, or the "preview" was removed.  "Figure 3 shows the predicted errors for the AIRS partial column XCH4 VMR within the pressure levels measured by the aircraft.  The measurement error (light green) is 18 ppb (from the last term of Eq. 7b) , and the total error for a single observation (including smoothing error) is 41 ppb.  A component of the total error, the cross-state error, is estimated as 21 ppb (from Eq. 7b)."  The smoothing error estimate now shown after the new Eq. 5 and the text states that this is calculated from Eq. 5.

Reviewer 2: (2) The authors use very often the term "partial column". They need to introduce how the partial methane column over the pressure range the aircraft had measured has been calculated, both for the aircraft and AIRS data (what shape of the methane profile? where do pressure and temperature profiles come from?); further, the figures usually show volume mixing ratios (in ppb) instead of partial colums. This might sound picky, but I think this accurateness in wording should be kept.

Response: At the beginning of Section 2, 2 sentences were added to describe how partial columns are created, "The retrieval estimates AIRS CH4 dry volume mixing ratio (VMR) profile. When a "partial column quantity" is validated, the retrieved CH4 profile is post-processed into partial column XCH4 VMR relative to dry air, with methodology from Connor et al. (2008) and Kulawik et al. (2017), where the VMR's at the pressure levels are weighted according to a pressure weighting function, resulting in a partial column volume mixing ratio (VMR).". Based on reviewer 2's comments here and elsewhere, the paper was previously not clear that the partial column quantity is XCH4 VMR. The wording was updated throughout the paper to indicate that the validated quantity was "the AIRS partial column XCH4 VMR within the pressure levels measured by the aircraft"

Reviewer 2 (3): I was a bit surprised that the two models that were used to assess the so-called "validation error" (i.e. the unknown stratospheric part of the profile) provided largely different results (4.4 vs. 15.7 ppb). I think the authors should elaborate a little more on the reasons for this large difference. This is particularly important because the averaging kernels in Fig. 2 demonstrates that the stratospheric information is large mapped into the troposphere below 300 hPa, i.e. the AIRS signal obviously depends a lot on the assumptions about the stratosphere. Is it possible that the use of other models leads to even larger estimates of the validation error?

Response:  Added additional text and reference on the errors from model extension of the aircraft profile: "The methane profile has a strong, spatially varying negative vertical gradient in the stratosphere.  Models in general have a positive bias in the extratropical stratosphere (Patra et al., 2011).  In GEOS-Chem 4x5, the column bias is shown in Figure 2c of Turner et al. (2015) and further discussed in Maasakkers (2019), which resolves the bias to the stratosphere, and model stratospheric accuracy is an active research area (Ostler et al., 2016; Maasakkers et al., 2019)."

Reviewer 2 (4) A side remark without relevance to the revision of this paper: the SPARC TUNER activity has worked on recommendations about error reporting. A paper on this topic has just been accepted by AMT. It would be nice to look into that paper (amt-2019-350) and possibly following these recommendations in future.

Response: Thank you for this information It is good to standardize validation.

Embedded notes/comments from Referee 2.  I summarized the comments and wrote the response.

1) Abstract: Define "validation error" Response: Update wording in abstract, and also change name to "validation uncertainty" "We estimate a 16 ppb validation uncertainty because the aircraft typically did not measure methane at altitudes where the AIRS measurements have some sensitivity, e.g. the stratosphere, and there is uncertainty in the truth that we validate against."

Section 2: Question about how the partial column is defined and the profile shape assumptions. The text was updated in response to this and to an earlier comment to clarify that a profile is retrieved, and then an XCH4 VMR is calculated. "The retrieval estimates AIRS CH4 dry volume mixing ratio (VMR) profile. When a "partial column quantity" is validated, the retrieved CH4 profile is post-processed into partial column XCH4 VMR relative to dry air, with methodology from Connor et al. (2008) and Kulawik et al. (2017), where the VMR's at the pressure levels are weighted according to a pressure weighting function, resulting in a partial column volume mixing ratio (VMR)."

3) cm-1 formatting. Response: Updated.

4) removed URL from citation as suggested by reviewer.

5) Request figure 1: center legend on Pacific. Update symbols. Remove gray color. Label aircraft sites. Response: Updated as requested

6) Description of Sa is needed. Response: Updated section around the new Equation 5, "The expected total error includes the smoothing error, which the covariance of the hcAccxc-x ac (Rodgers, 2000), where the covariance of xc-x ac is the a priori covariance, Saxx. The smoothing error is: (Eq 5) We estimate the smoothing error for the partial column XCH4 VMR within the pressure levels measured by the aircraft to be 30 ppb, using Eq. 5. This estimate strongly depends on Saxx, the a priori covariance, which is the same as in Worden et al. (2012); briefly 5% diagonal variability with correlations in pressure set from the MOZART model."

7) Description of the selection of quality flags. "The specific flags used for AIRS CH4 are as follows, which were set by minimizing the standard deviation of small clusters of retrievals and to standardize the sensitivity:"

8) The reviewer suggests that things like "|KdotdL| < 0.23 should be given proper names. Response: Since these variables are not referred to outside of this section, and additionally are referencing an existing data user's guide, the updates were to the description of the equation, e.g. |KdotdL| < 0.23 --> The absolute value of KdotdL < 0.23.

Equation 1, question about "G" versus "Gr". Response: Removed the R. Thanks.

Question about "sum over i" in Equation 1. Response: change b to a vector and the summation is implied just like all the other matrix multiplication.

Comment on A = dx/dx "Careful with log! Response: Yes, thank you, the equations are mixing up x and log(x). The intention was that x=log(VMR), so A = dx/dx, and K = dradiance/dx = dradiance/d(log(VMR)). Other issues fixed in this section from the reviewer.

Reviewer: Comment on estimate of smoothing error. Response: Addressed above, the smoothing error was estimated from the new Eq. 5.

Reviewer at Eq. 5b, "It is necessary to mention that this step is possible because the levels of the aircraft measurement are assumed uncorrelated." Response: This is now Eq. 6a, and the text was updated, "Equation 6a accounts for all of the AIRS smoothing error, whereas Equation 6b (the equation used in this work, other than Section 3.3) only accounts for the smoothing error from the part of the atmosphere measured by the aircraft profile. [...] The difference from Eqs. 6a and 6b is discussed in Section 3.3. "

Eq 6, "This step comes as surprise. Including 1-2 steps or explanation would be helpful." Response: Added new Eq. 5, introducing the a priori covariance, Eq. 7a, introducing the bias, and Eq. 7b, the covariance of 7a. Updated wording, "Equation 7a is the predicted bias between x_c (the measured AIRS value) and x_aircraft_c (the aircraft value with the AIRS Averaging kernel applied), and is the expected difference of Eqs. 4 and 6b. Equation 7b is the covariance of Eq. 7a, and estimates the predicted error."

Equation 6, "Acb is not defined". Response: change Acb to GcKb.

Line 235, "Here the explanation should be given that this is because the retrieval vector is log(VMR)." Response: updated. "Because the retrieved quantity log(VMR), the error in ppb is approximately the fractional error times the methane value in ppb."

Line 255, "Define validation error". Response: Now it is consistently called "validation uncertainty" throughout the paper rather than "validation error" and "validation uncertainty".

Line 259, "Where do these errors come from?". Response: These errors are now tied to equations.

Line 266, The reviewer notes that not only the stratosphere influences the troposphere, but the troposphere influences the stratosphere, due to the broad sensitivity. Response: Added sentence, "Similarly, the true state in the troposphere influences retrieved values in the stratosphere."

Line 282. Is the 16 ppb the larger of the two errors from model propagation? Response: We updated this to set the "validation uncertainty" to the average of the two model results, 10 ppb. This agrees with the previous result from Wunch et al. (2010).

Reviewer comment: How does the profile extension relate to the "validation uncertainty"? Doesn't it relate more to the bias estimate? Response: The profile extension obviously affects the bias versus validation data which affects the bias correction. However, it is most relevant to the uncertainty that results from extending the aircraft profile using a model. Text update, "Appendix A shows further analysis of mean differences of AIRS minus aircraft for different profile extension choices. The bias varies by ~ 5 ppb for different profile extension choices when comparing at 700 hPa, ~10 ppb for different profile extension choices when comparing at 500 hPa, and ~11 ppb for different profile extension choices when comparing the column above 750 hPa."

Line 291. Reviewer requests that Worden et al., 2011 bias correction be summarized briefly. Response: Added text, "We therefore use the bias correction approach described in Worden et al. (2011), where a bias profile (which varies by pressure) is passed through the averaging kernel to account for the AIRS sensitivity, as seen in Eq. 8. The form of the bias profile, delta_bias set by Eq. 9."

Eq 7. Reviewer wonders if A is a vector. Response: A is a matrix, as x-hat is a profile, as shown in Fig. 6. Text updated, "Where x = ln(VMR), because the retrieved quantity is ln(VMR), delta_bias is a vector, and A is the averaging kernel matrix for x = ln(VMR)"

Eq 7. Reviewer wonders about the handling of log. Response: text and equations updated to handle log consistently.

Line 315. Reviewer wonders why there is a bias peak about 200 hPa. Response: It seems that the bias is increasing through the stratosphere, whereas the sensitivity is decreasing above 100 hPa, and these two combine to generate a peak about 200 hPa. But this is speculation.

Section 4 title changed from "Evaluation against aircraft data by latitude" to "Evaluation against aircraft data versus latitude", because as the reviewer points out, there is no fitting of bias or skill as a function of latitude.

Line 319 Reviewer suggests wording update to, "Figure 6 shows a comparison between all AIRS measurements within 50 km and 9h of an aircraft measurement over the pressure range of the partial column measured by the aircraft.". Response: The wording was updated to, "Figure 5 [figure numbering update] shows a comparison between all AIRS measurements within 50 km and 9h of an aircraft measurement and the aircraft. The quantity compared is the partial column XCH4 VMR within the pressure levels measured by the aircraft."

Figure 7 legend unreadable. Response: Add text in the figure caption with the information in the legend.

Line 334-335, reviewer questions what the legend means, e.g. 0.0 +/- 4. Response: This was not well explained in the text previously. This is an estimate of the overall bias and uncertainty in the bias. Each campaign or station's bias is independently calculated and then the mean and standard deviation are calculated. A new sub-section was added at the end of Section 4.1, "4.2.6 The bias and bias uncertainty The bias is estimated by calculating the mean bias for each campaign or station separately, then calculating the mean and standard deviation for all campaigns / stations. The bias versus HIPPO is $0 \pm 4$ ppb. The bias versus ATom is $3 \pm 4$ ppb. The bias versus NOAA measurements is $9 \pm 7$ ppb."

Reviewer wants "roughly the same location and time" defined in line 344. Response: Text added to clarify, "The bias component is approximately the same for all AIRS methane measurements taken on the same day within 50 km, as we do not expect large variations in temperature and water vapor errors over these scales, which we presume to be a driver of these correlated errors."

Reviewer says this sentence is hard to parse, "The number of AIRS observations averaged ranges from 9 to 53 and averages 20.". Response: This was updated to, "We average over 1 day, the AIRS observations matching a single HIPPO or ATom measurement, within +-50 km and 9 hours of the measurement. We specify that there needs to be at least 9 AIRS observations for each comparison so that the systematic error, and not the precision (or measurement error), is the dominant term. These daily AIRS averages contain, on average, 20 AIRS observations."

In Figure 9, the reviewer noted that the figure showed VMR but was indicated as a partial column. Response: This is an XCH4 partial column, as noted above and now explained in the paper.` The wording describing the partial column throughout the paper, to remove this confusion, is, "The partial column XCH4 VMR within the pressure levels measured by the aircraft".

365-375. The reviewer says that this section is hard to follow and needs explanation on the importance of the values shown. Response: Updated this text to indicate what the numbers mean, and what the findings are. Also separate into 4 new subsections, discussing daily, monthly, 3-month, and seasonal cycle averaging. Added a new section on the bias characterization (in response to a previous comment).

The rewritten text is (the next 7 paragraphs):

[revised manuscript text omitted]

$$\hat{x} = \mathbf{h}x_{a} + \mathbf{h}\Lambda_{xx}(x - x_a) + \mathbf{h}G_R\sum_i K_i^b(b_i - b_i^a) + \mathbf{h}\Lambda_{xy}(y - y_a) + \mathbf{h}Gn \tag{3b}$$

$$\hat{x} = hx_a + hA_{xx}(x - x_a) + hG_x K^b b_{error} + hA_{xy}(y - y_a) + hG_x n \tag{3b}$$

In Eq. 3a, the vector $\hat{\mathbf{x}} \quad \hat{x}$ (denoted in bold) is converted to the scalar of interest, $\hat{x}$ (non-bold, italic). In our validation comparisons, $\mathbf{h}$ is used to select 1) a specific pressure level that is measured by the aircraft, 2) the partial column XCH$_4$ VMR within the pressure levels measured by the aircraft, and 3) the partial column XCH$_4$ between 750 hPa and the top of the atmosphere.

**3.2 Approach for Comparing AIRS measurements to aircraft profiles**

A challenge in comparing the satellite-based AIRS measurements to aircraft data is that the aircraft will typically measure only a section of the atmosphere (e.g. the troposphere), whereas the AIRS measurements are sensitive, to varying degrees (see Fig. 2), to the entire atmosphere. To account for these differences, we divide the atmosphere into two parts $\mathbf{x} = [\mathbf{x}_c, \mathbf{x}_s]$: where $\mathbf{x}_c$ is the part measured by the aircraft (denoted $\mathbf{c}$ for airCraft), and $\mathbf{x}_s$ is the part not measured by the aircraft (denoted $\mathbf{s}$ for Stratospheric):

$$\hat{x}_c = \mathbf{h}_c x_a + \mathbf{h}_c \Lambda_{cc}(x_c - x_a^c) + \mathbf{h}G_R\sum_i K_i^b(b_i - b_i^a) + \mathbf{h}_c\Lambda_{cy}(y - y_a) + \mathbf{h}_c\Lambda_{cs}(x_s - x_a^s)\Lambda_{cs}(x_s - x_a^s) + \mathbf{h}_c Gn \tag{4}$$

$$\hat{x}_c = h_c x_a + h_c A_{cc}(x_c - x_a^c) + hG_c K^b b_{error} + h_c A_{cy}(y - y_a) + h_c A_{cs}(x_s - x_a^s)A_{cs}(x_s - x_a^s) + h_c G_c n \tag{4}$$

where the term $\mathbf{A_{cs}}$ is the cross-term in the averaging kernel that describes the partial derivatives of the aircraft-measured levels (e.g. the troposphere) to the un-measured levels (e.g. the stratosphere). Equation 4 describes how the AIRS measurement $\hat{x}_c \hat{x}_c$ responds to the true state $[\mathbf{x}_c, \mathbf{x}_s]$. So, if for example, the aircraft measured indices [0:9], and did not measure pressure levels [10:*], then $A_{cc} = A[0:9, 0:9]$ and $A_{cs} = A[0:9, 10:65]$, where $A$ is the full averaging kernel.

We compare our AIRS observation, $\hat{x}_c \hat{x}_c$ in Eq. 4, to our aircraft observation, $x_{aircraft} x_{aircraft}$. To compare directly to the aircraft observation (without accounting for AIRS sensitivity) we would compare to $\hat{x}_{aircraft}^c = \mathbf{h}_c x_{aircraft}$. However, the $\hat{x}_{aircraft}^c = h_c x_{aircraft}$. The expected total error  includes the smoothing error, which

ppb.  the covariance of the $h_c A_{cc}(x_c - x_a^c)$ (Rodgers, 2000), where the covariance of $(x_c - x_a^c)$ is the a priori covariance, $S_a^{xx}$. The smoothing error is:

$$\hat{x}_{aircraft}^{\epsilon} = h_c x_a^{-} + h_c A_{cc}\left(x_{aircraft}^{\epsilon} - x_a^{\epsilon}\right) + h_c A_{cs}\left(x_{aircraft}^{s} - x_a^{s}\right) \tag{5a}$$

$$\text{Smoothing error} = h_c A_{cc} S_a^{xx} A_{cc}^T h_c^T \tag{5}$$

We estimate the smoothing error for the partial column XCH$_4$ VMR within the pressure levels measured by the aircraft to be 30 ppb, using Eq. 5. This estimate strongly depends on $S_a^{xx}$, the a priori covariance, which is the same as in Worden et al. (2012); briefly 5% diagonal variability with correlations in pressure set from the MOZART model. In Equation 6a, we apply the AIRS Averaging kernel to the aircraft measurement to fully account for the AIRS sensitivity:

$$\hat{x}_{aircraft}^{c} = h_c x_a + h_c A_{cc}\left(x_{aircraft}^{c} - x_a^{c}\right) + h_c A_{cs}\left(x_{aircraft}^{s} - x_a^{s}\right) \tag{6a}$$

$$\hat{x}_{aircraft}^{c} = h_c x_a + h_c A_{cc}\left(x_{aircraft}^{c} - x_a^{c}\right) \tag{6b}$$

One issue is that we do not actually have aircraft observations in the "s" part of the atmosphere, $ x_{aircraft}^{s}$, which is used in the second term of Eq. 6a. We have aircraft observations in the "c" part of the atmosphere only, so we apply the Averaging Kernel to this part of the atmosphere only

$$\hat{x}_{aircraft}^{\epsilon} = h_c x_a^{-} + h_c A_{cc}\left(x_{aircraft}^{\epsilon} - x_a^{\epsilon}\right) \tag{5b}$$

. Equation 6a accounts for all of the AIRS smoothing error, whereas Equation 6b (the equation used in this work, other than Section 3.3) only accounts for the smoothing error from the part of the atmosphere measured by the aircraft profile. The difference from Eqs. 6a and 6b is discussed in Section 3.3.

 Equation 7a is the predicted bias between $ \hat{x}_c$ (the measured AIRS value) and $ \hat{x}_{aircraft}^{c}$, (the aircraft value with the AIRS Averaging kernel applied), and is the expected difference of Eqs. 4 and

$$E|||| = h_c\left(A_{cb}S_a^{bb}A_{cb}^T + A_{cy}S_a^{yy}A_{cy}^T + A_{cs}S_a^{ss}A_{cs}^T + S_m^{\epsilon\epsilon}\right)h_c^T \tag{6}$$

The matter $S_a$ term describes the *a priori* uncertainty of methane, interferents, or systematic parameters, which propagate into the 6b. Equation 7b is the covariance of Eq. 7a, and estimates the predicted error in the first 3 terms::

$$E(\hat{x}_c - \hat{x}^c_{aircraft}) = h_c G_c K^b E(b_{error}) + h_c A_{cy} E(\hat{y} - y_a) + h_c A_{cs} E(\hat{x}_s - x_s) + h_c G_c E(n_{error}) \qquad \textbf{(7a)}$$

$$E||\ (\hat{x}_c - \hat{x}^c_{aircraft})|| = h_c (G_c K^b S^{bb}_a K^T_b G^T_c + A_{cy} S^{yy}_a A^T_{cy} + A_{cs} S^{ss}_a A^T_{cs} + S^{cc}_m) h^T_c \qquad \textbf{(7b)}$$

Eq. 7a represents the propagation of mean biases from: (1) $\mathbf{A_{cb} S^{bb}_a A^T_{cb}}$ describes systematic error non-retrieved parameters and assumptions, 
[revised manuscript text omitted]

---

## Author Response (AR2)

"Legends and axes in some figures seem to be out of alignment in the manuscript PDF file. If the authors need to modify the original figures, they should deal with these technical problems before the publication."

Response:
I found and fixed: figure 3 was cutoff at the top and bottom, figure 5 & 7 the title was not centered, the panel labels on Fig 9 were not aligned.  I don't see any other issues in the figures.